# Devil is in the Details: Density Guidance
# for Detail-Aware Generation with Flow Models

**Rafał Karczewski** [1]  **Markus Heinonen** [1]  **Vikas Garg** [1 2]

## Abstract

Diffusion models have emerged as a powerful class of generative models, capable of producing high-quality images by mapping noise to a data distribution. However, recent findings suggest that image likelihood does not align with perceptual quality: high-likelihood samples tend to be smooth, while lower-likelihood ones are more detailed. Controlling sample density is thus crucial for balancing realism and detail. In this paper, we analyze an existing technique, Prior Guidance, which scales the latent code to influence image detail. We introduce score alignment, a condition that explains why this method works and show that it can be tractably checked for any continuous normalizing flow model. We then propose Density Guidance, a principled modification of the generative ODE that enables exact log-density control during sampling. Finally, we extend Density Guidance to stochastic sampling, ensuring precise log-density control while allowing controlled variation in structure or fine details. Our experiments demonstrate that these techniques provide fine-grained control over image detail without compromising sample quality. Code is available at https://github.com/Aalto-QuML/density-guidance.

## 1. Introduction

Diffusion models are a family of generative models that learn to map noise to a data distribution $p_0$, which allows realistic image sampling (Ho et al., 2020; Song et al., 2021b;a; Vahdat et al., 2021). In the quest towards high-fidelity sampling it is natural to ask whether perceptual quality of images aligns with their likelihood $p_0(\boldsymbol{x})$ (Karczewski et al., 2025)?

[1]Department of Computer Science, Aalto University, Finland [2]YaiYai Ltd. Correspondence to: Rafał Karczewski <rafal.karczewski@aalto.fi>.

*Proceedings of the 42$^{nd}$ International Conference on Machine Learning*, Vancouver, Canada. PMLR 267, 2025. Copyright 2025 by the author(s).

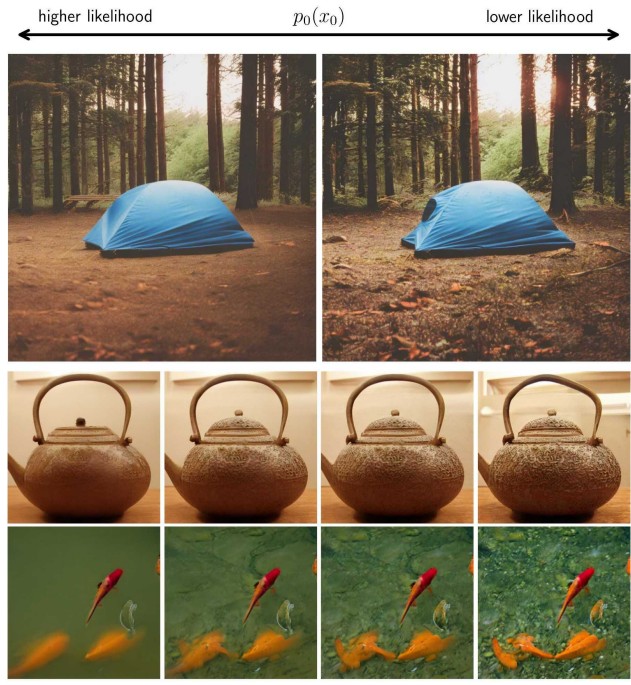

higher likelihood  $p_0(x_0)$  lower likelihood

*Figure 1.* **Density guidance controls the amount of detail.** Images sampled from the same condition and latent code with different strengths of guidance. Top: StableDiffusion v2.1 (Rombach et al., 2022). Bottom: EDM2 (Karras et al., 2024b).

Remarkably, the density $p_0$ correlates *negatively* with the amount of detail: within the *typical samples* (Nalisnick et al., 2019) $\boldsymbol{x} \sim p_0$ higher-density images tend to lack detail and be smooth, while lower-density images tend to be richly textured and detailed (Sehwag et al., 2022) (See Fig. 1). Outside the typical samples, extremely low density leads to broken images (Karras et al., 2024a), while extremely high density strips detail to the point of resembling sketch drawings or blurs (Karczewski et al., 2025).

Somewhat surprisingly the common sampling strategies in flow models do not optimise for sample density (Karras et al., 2022). Recently, Karczewski et al. (2025) proposed an approach towards controlling the sample density by biasing the sampling towards the extremely high likelihood regions of $p(\boldsymbol{x}_0|\boldsymbol{x}_t)$, and demonstrated that these correspond to unrealistic images. Their work is limited in three ways.

The approach is only derived for SDE models with linear drift. The exact procedure is not tractable so the authors resort to approximations in practice. Finally, it only allows targeting the highest possible likelihoods, which do not produce realistic images. This highlights the need for a more general approach that allows fine-grained control over sample density while preserving realism.

In this paper, we build upon prior observations that scaling the latent code affects image detail (Song et al., 2021b). We refer to this method as *Prior Guidance* and we provide a theoretical explanation for this phenomenon by introducing *score alignment*, a condition under which Prior Guidance provably increases or decreases log-density. We show that this condition often holds in practice.

Beyond this analysis, we introduce *Density Guidance*, a novel procedure that allows explicit control over the log-density of generated samples. Assuming knowledge of the score function, we derive an alternative ODE that guarantees the log-density of the trajectory evolves exactly as specified. Empirically, we show that this method achieves similar results to Prior Guidance.

Finally, we extend Density Guidance to incorporate stochastic sampling. This enables precise control over the log-density of generated samples even when randomness is introduced. By injecting noise at different stages of the generation process, we can selectively influence variations in high-level structure (e.g., shape and composition) or fine-grained details. Our experiments demonstrate that this stochastic extension allows for enhanced diversity while preserving control over the desired level of detail.

In summary, in this paper we

- introduce *Score Alignment*, a condition that explains how latent code scaling affects image detail and can be tractably checked for any CNF model - section 3;

- derive a modification of the generative ODE that enables exact log-density control during sampling - *Density Guidance* - section 4;

- extend Density Guidance to stochastic sampling, retaining exact log-density control while allowing controlled variation in structure or details - section 5.

## 2. Background

Let $\boldsymbol{x} \in \mathbb{R}^D$. We assume spatial gradient $\nabla = (\frac{\partial}{\partial x_1}, \ldots, \frac{\partial}{\partial x_D})^T \in \mathbb{R}^D$, divergence $\mathrm{div} = \sum_d \frac{\partial}{\partial x_d} \in \mathbb{R}$, and Laplacian $\Delta = \sum_d \frac{\partial^2}{\partial x_d^2} \in \mathbb{R}$ operators. We assume continuous time $t \in [0, T]$ between data $p_0$ and noise $p_T$.

### 2.1. Continuous normalizing flows

Continuous normalizing flows (CNFs) (Chen et al., 2018) are probabilistic models specified by a tractable prior distribution $p_T$ at terminal time $T$ and an ordinary differential equation (ODE)

$$d\boldsymbol{x}_t = \boldsymbol{u}_t(\boldsymbol{x}_t)dt, \tag{1}$$

which samples by integrating from $\boldsymbol{x}_T \sim p_T$ at $t = T$ to $t = 0$ by following the time-varying vector field $\boldsymbol{u}_t : \mathbb{R}^D \mapsto \mathbb{R}^D$ with a solution

$$\boldsymbol{x}_t := \boldsymbol{x}_t(\boldsymbol{x}_T) = \boldsymbol{x}_T + \int_T^t \boldsymbol{u}_\tau(\boldsymbol{x}_\tau)d\tau. \tag{2}$$

The flow family encompasses many popular generative frameworks, including diffusion/score-based models (Song et al., 2021b), flow matching (Lipman et al., 2023; Tong et al., 2024), rectified flows (Liu et al., 2023), stochastic interpolants (Albergo & Vanden-Eijnden, 2023), consistency models (Song et al., 2023), and the denoising probabilistic models at continuous limit (Kingma et al., 2021): the vector field $\boldsymbol{u}_t$ is the denoiser.

In a CNF we can evaluate the log-likelihood of a sample moving according to Eq. 1 with (Chen et al., 2018):

$$\frac{d \log p_t(\boldsymbol{x}_t)}{dt} = -\mathrm{div}\, \boldsymbol{u}_t(\boldsymbol{x}_t), \tag{3}$$

where $p_t$ is the marginal density of a process defined by Eq. 1. Karczewski et al. (2025); Skreta et al. (2025) generalized this formula to enable tracking of the marginal $p_t$ for a sample following a *different* direction $d\boldsymbol{x}_t = \tilde{\boldsymbol{u}}_t(\boldsymbol{x}_t)dt$ as

$$\frac{d \log p_t(\boldsymbol{x}_t)}{dt} = -\mathrm{div}\, \boldsymbol{u}_t(\boldsymbol{x}_t) \tag{4}$$
$$+ \nabla \log p_t(\boldsymbol{x}_t)^T \Big( \tilde{\boldsymbol{u}}_t(\boldsymbol{x}_t) - \boldsymbol{u}_t(\boldsymbol{x}_t) \Big).$$

At $\tilde{\boldsymbol{u}}_t = \boldsymbol{u}_t$, this reduces back to Eq. 3. See Appendix B for detailed derivations.

### 2.2. Diffusion models

A notable case of flow models are diffusion models given by a forward process $p_t(\boldsymbol{x}_t|\boldsymbol{x}_0) = \mathcal{N}(\alpha_t \boldsymbol{x}_0, \sigma_t^2 \boldsymbol{I}_D)$, or equivalently, by a stochastic differential equation (SDE)

$$d\boldsymbol{x}_t = f(t)\boldsymbol{x}_t dt + g(t)d\mathrm{W}_t \tag{5}$$

with drift $f(t) = \frac{d \log \alpha_t}{dt}$, diffusion $g^2(t) = 2\sigma_t^2 \frac{d \log \frac{\sigma_t}{\alpha_t}}{dt}$, and Wiener process $\mathrm{W}_t$. A CNF with drift

$$\boldsymbol{u}_t^{\text{PF-ODE}}(\boldsymbol{x}_t) = f(t)\boldsymbol{x}_t - \frac{1}{2}g^2(t)\underbrace{\nabla \log p_t(\boldsymbol{x}_t)}_{\text{score}} \tag{6}$$

shares marginals $p_t$ with Eq. 5 when $p_T$ are shared (Song et al., 2021b). This Probability-Flow ODE (PF-ODE) is an efficient, deterministic, sampler in diffusion models.

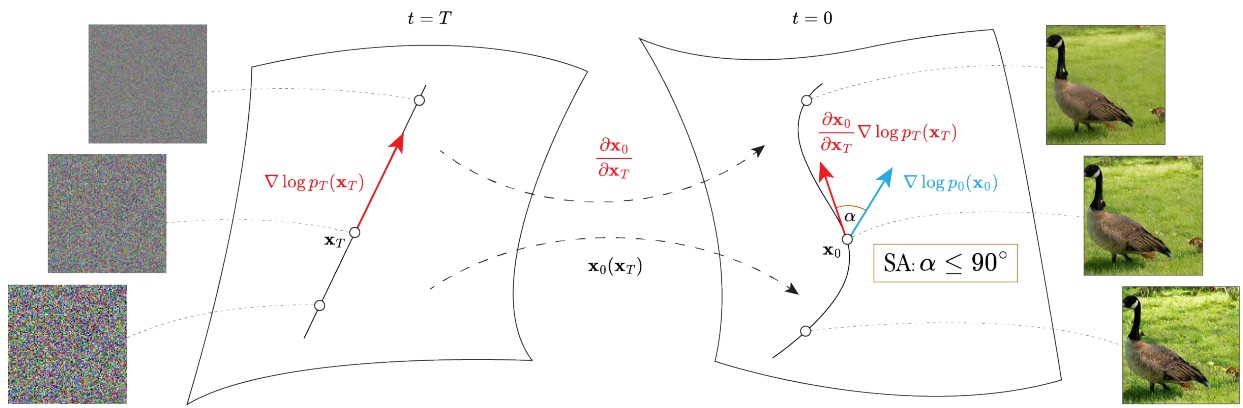

*Figure 2.* **Prior Guidance and Score Alignment (SA).** Prior guidance works by moving $\boldsymbol{x}_T$ (noise) in the direction of $\nabla \log p_T(\boldsymbol{x}_T)$ and decoding to $\boldsymbol{x}_0$ (image). The purpose of this is to increase or decrease $\log p_0(\boldsymbol{x}_0)$, which is inversely related to the level of detail in $\boldsymbol{x}_0$. SA is a condition that ensures prior guidance is effective by requiring the alignment of score vectors across time steps. Red arrows represent tangents to the curves: $\nabla \log p_T(\boldsymbol{x}_T)$ is the tangent to the left curve at $\boldsymbol{x}_T$, and its push-forward via $\frac{\partial \boldsymbol{x}_0}{\partial \boldsymbol{x}_T}$ is the tangent to the decoded curve at $\boldsymbol{x}_0$. SA states that the transformed tangent vector must align with $\nabla \log p_0(\boldsymbol{x}_0)$ such that the angle $\alpha \leq 90°$ (non-negative dot product).

## 2.3. Stochastic sampling and likelihood

Eijkelboom et al. (2024) pointed out that any CNF can be cast as an SDE model via the score function $\nabla \log p_t(\boldsymbol{x})$:[1]

$$d\boldsymbol{x}_t = \Big(\boldsymbol{u}_t(\boldsymbol{x}_t) - \frac{1}{2}\varphi^2(t)\nabla \log p_t(\boldsymbol{x}_t)\Big)dt + \varphi(t)d\overline{\mathbf{W}}_t, \quad (7)$$

where $\overline{\mathbf{W}}$ is the Wiener process going backward in time and the process defined by Eq. 7 shares marginals with Eq. 1 for any choice of the variance term $\varphi$ as long as they share $p_T$. Karczewski et al. (2025) demonstrated that for SDE models one can also track the evolution of marginal

$$d \log p_t(\boldsymbol{x}_t) = F(t, \boldsymbol{x}_t)dt + \varphi(t)\nabla \log p_t(\boldsymbol{x}_t)^T d\overline{\mathbf{W}}_t, \quad (8)$$

where

$$\begin{aligned} F(t, \boldsymbol{x}) = &- \operatorname{div} \boldsymbol{u}_t(\boldsymbol{x}) - \frac{1}{2}\varphi^2(t)\Delta \log p_t(\boldsymbol{x}) \\ &- \frac{1}{2}\varphi^2(t)\|\nabla \log p_t(\boldsymbol{x})\|^2. \end{aligned} \quad (9)$$

## 2.4. Neural network approximations

In all our experiments, we assume access to pre-trained neural networks to approximate $\mathbf{u}_t$ and $\nabla \log p_t$. When estimating Jacobian-vector products such as $\frac{\partial \boldsymbol{u}_t}{\partial \boldsymbol{x}}\boldsymbol{v}$, we leverage automatic differentiation to compute these efficiently with a single network pass. To estimate divergence, such

as $\Delta \log p_t(\boldsymbol{x}) = \operatorname{div} \nabla \log p_t(\boldsymbol{x})$, we use the Hutchinson's trick (Hutchinson, 1989; Grathwohl et al., 2019) using a single Rademacher test vector.

## 3. Scaling the latent code – when and why?

In this section we evaluate whether the simple latent rescaling approach of Song et al. (2021b) is sufficient to control for sample density, and demonstrate its shortcomings.

Song et al. (2021b) observed that scaling the latent noise $\boldsymbol{x}_T \sim \mathcal{N}(\mathbf{0}, \mathbf{I})$ down with $\beta < 1$ monotonically decreases the amount of detail in the decoded images $\boldsymbol{x}_0^{\text{PF-ODE}}(\beta\boldsymbol{x}_T)$, while preserving the image semantics (See Fig. 2). Interestingly, Karczewski et al. (2025) recently showed that the log-density $\log p_0(\boldsymbol{x}_0)$ assigned by a diffusion model correlates with the amount of detail in the generated image. Most diffusion models use a Gaussian noise $p_T = \mathcal{N}(\mathbf{0}, \boldsymbol{I}_D)$, whose score $\nabla \log p_T(\boldsymbol{x}_T) = -\boldsymbol{x}_T$ simply reduces the latent norm towards zero. Thus (down)scaling at $t = T$ is equivalent to maximizing $\log p_T$.

These two observations suggest a simple hypothesis:

> **Prior guidance**: To increase (decrease) $\log p_0(\boldsymbol{x}_0)$, it suffices to move $\boldsymbol{x}_T$ in the positive (negative) direction of $\nabla \log p_T(\boldsymbol{x}_T)$, and then decode.

We are interested in studying whether (steepest) $\log p_T$ increase in latent $\boldsymbol{x}_T$ leads to a monotonic increase in $\log p_0$ of the decoding $\boldsymbol{x}_0(\boldsymbol{x}_T)$ Eq. 2. To formalise this notion, we assume a latent curve $c : [0, 1] \to \mathbb{R}^D$ at $t = T$, whose tangent is given by the score $c'(s) = \nabla \log p_T(c(s))$. A

---

[1]In Eijkelboom et al. (2024), the drift is: $\boldsymbol{u}_t(\boldsymbol{x}_t) + \frac{1}{2}\varphi^2(t)\nabla \log p_t(\boldsymbol{x}_t)$. This is caused by different conventions. We follow the diffusion literature (Song et al., 2021b), where time flows backward, i.e. $dt < 0$ and the Wiener process runs in reverse during sampling. In Eijkelboom et al. (2024) sampling is from $t = 0$ to $t = 1$ and $dt > 0$.

monotonic curve decoding has

$$\frac{d}{ds}\log p_0\Big(\boldsymbol{x}_0\big(c(s)\big)\Big) \geq 0, \qquad \forall s, \qquad (10)$$

which is equivalent to *Score Alignment* (Appendix C):

$$\text{SA}: \quad \underbrace{\nabla \log p_0(\boldsymbol{x}_0)^T}_{\text{decoding score}} \underbrace{\frac{\partial \boldsymbol{x}_0}{\partial \boldsymbol{x}_T} \nabla \log p_T(\boldsymbol{x}_T)}_{\text{push-forward score } \boldsymbol{v}_0 \in \mathbb{R}^D} \geq 0, \quad (11)$$

for all $\boldsymbol{x}_T \in \mathbb{R}^D$, where $\boldsymbol{v}_t(\boldsymbol{x}_T) \coloneqq \frac{\partial \boldsymbol{x}_t}{\partial \boldsymbol{x}_T} \nabla \log p_T(\boldsymbol{x}_T)$. In Appendix C.2 we show that Eq. 11 always holds when $\boldsymbol{u}_t$ is linear in $\boldsymbol{x}$, as in trivial diffusion models. We also show that it does not hold in general by providing a counterexample. See Fig. 2 for a visualisation.

To evaluate the SA Eq. 11 we need to solve for $\boldsymbol{v}_0$. We use sensitivity equations (i.e. forward differentiation)

$$d \begin{bmatrix} \boldsymbol{x}_t \\ \boldsymbol{v}_t \end{bmatrix} = \begin{bmatrix} \boldsymbol{u}_t(\boldsymbol{x}_t) \\ \frac{\partial \boldsymbol{u}_t(\boldsymbol{x}_t)}{\partial \boldsymbol{x}} \boldsymbol{v}_t \end{bmatrix} dt, \qquad (12)$$

with $\boldsymbol{v}_T \coloneqq \nabla \log p_T(\boldsymbol{x}_T)$ to describe the $\boldsymbol{v}_t$ evolution, and specifically, to solve $\boldsymbol{v}_0(\boldsymbol{v}_T)$ (Baydin et al., 2018).

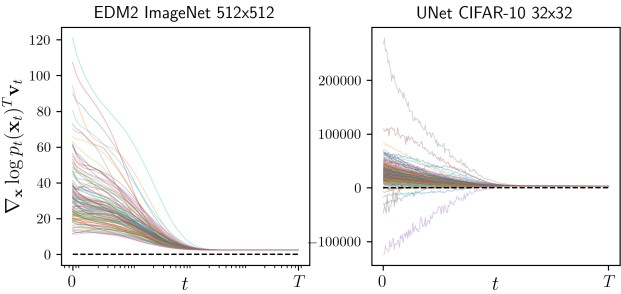

Figure 3. **Nearly all $\boldsymbol{x}_T$ satisfy the positive score alignment of Eq. 11 across models and datasets.**

**Empirical demonstration** To empirically verify whether SA holds, one can sample a large batch of $\boldsymbol{x}_T \sim p_T$ and solve Eq. 12 from $t = T$ to $t = 0$[2] and check whether $\boldsymbol{v}_0^T \nabla \log p_0(\boldsymbol{x}_0) \geq 0$. We demonstrate this on two models, a VP-SDE model trained on CIFAR-10 (Karczewski et al., 2025), and EDM2, a conditional latent diffusion trained on ImageNet512 (Karras et al., 2024b). We find that for CIFAR 97% of the latent codes satisfy the equation and 100% for EDM2 (Fig. 3). This shows that, in most cases, scaling the latent code $\boldsymbol{x}_T$ impacts $\nabla \log p_0(\boldsymbol{x}_0)$ monotonically, and thus explains the visual effect of low-level feature manipulation (Fig. 1). See Appendix M for more samples.

**Log-density vs FLIPD** Kamkari et al. (2024) recently proposed FLIPD - a method for measuring local intrinsic dimension and argued that it correlates strongly with

---

[2]In practice, we solve until $t = \varepsilon$ for small $\varepsilon > 0$.

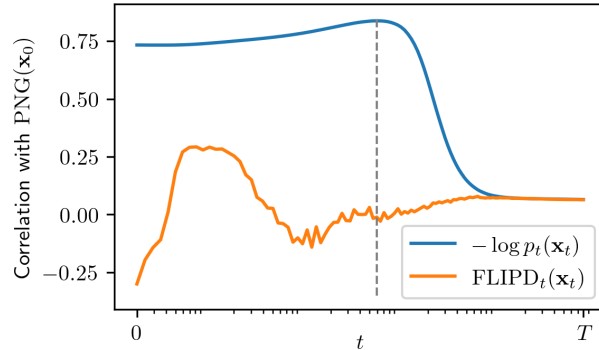

Figure 4. **Negative $\log p_t(\boldsymbol{x}_t)$ correlates well with image compression size**, while the recently proposed intrinsic dimensionality measure FLIPD (Kamkari et al., 2024) correlates weakly. Experiment performed for a latent diffusion model EDM2 trained on ImageNet $512 \times 512$ resolution (Karras et al., 2024b).

the amount of detail (or information) in the image as measured by the size of PNG compression of the decoded image $\text{PNG}(\boldsymbol{x}_0)$. In Fig. 4 we show on a high resolution latent diffusion model EDM2 (Karras et al., 2024b) that $-\log p_t(\boldsymbol{x}_t)$ correlates with $\text{PNG}(\boldsymbol{x}_0)$ more strongly, reaching a maximum of 84%, compared to 29% achieved by FLIPD. Furthermore, we observed that the correlation of $-\log p_t(\boldsymbol{x}_t)$ with $\text{PNG}(\boldsymbol{x}_0)$ is the strongest not for $t \approx 0$, but rather $t \approx 0.6$ corresponding to $\log \text{SNR}(t) = \log \frac{\alpha_t^2}{\sigma_t^2} = 1$. This suggests that for detail manipulation with Prior Guidance, verification of the SA condition Eq. 11 should be done up to this value of $t$ rather than $t = 0$.

**What if score is unknown?** To verify SA, one needs to solve Eq. 12 and estimate $\nabla \log p_0(\boldsymbol{x}_0)$. The latter is straightforward for diffusion models, but not in the general case of Eq. 1. Remarkably, it is also possible to evaluate $\boldsymbol{v}_0^T \nabla \log p_0(\boldsymbol{x}_0)$ for any flow model without estimating the score itself. Concretely, in Appendix C.1, we show that for $\omega_t \coloneqq \boldsymbol{v}_t^T \nabla \log p_t(\boldsymbol{x}_t)$, $\frac{d}{dt}\omega_t = -\operatorname{div}(\frac{\partial \boldsymbol{u}_t}{\partial \boldsymbol{x}}(\boldsymbol{x}_t)\boldsymbol{v}_t)$. Therefore, in absence of the score function, one can estimate $\omega_0$ by augmenting Eq. 12 with $\dot{\omega}_t$ initialized at $\omega_T = \|\nabla \log p_T(\boldsymbol{x}_T)\|^2$:

$$d \begin{bmatrix} \boldsymbol{x}_t \\ \boldsymbol{v}_t \\ \omega_t \end{bmatrix} = \begin{bmatrix} \boldsymbol{u}_t(\boldsymbol{x}_t) \\ \frac{\partial \boldsymbol{u}_t(\boldsymbol{x}_t)}{\partial \boldsymbol{x}} \boldsymbol{v}_t \\ -\operatorname{div}\left(\frac{\partial \boldsymbol{u}_t(\boldsymbol{x}_t)}{\partial \boldsymbol{x}} \boldsymbol{v}_t\right) \end{bmatrix} dt. \qquad (13)$$

In Fig. 6 we present the SA verification algorithm. To empirically validate Eq. 13, we used a VP-SDE CIFAR-10 diffusion model (Karczewski et al., 2025), sampled 256 latent codes $\boldsymbol{x}_T$ and solved Eq. 13 from $t = T$ to $t$ corresponding to $\log \text{SNR}(t) = 1$. This is a score-based model and thus we can compare the ground truth $\boldsymbol{v}_t^T \nabla \log p_t(\boldsymbol{x}_t)$ with the estimated $\omega_t$. We found that their correlation was at 98.8%. See Fig. 5.

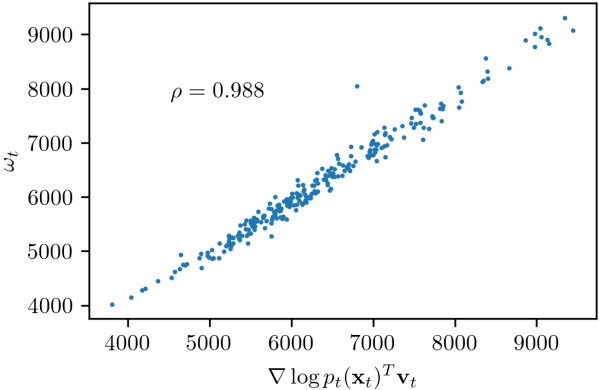

*Figure 5.* **The augmented sensitivity equations of Eq. 13** accurately tracks the score alignment (SA).

# 4. Density Guided Sampling

In section 3 we discussed ways to determine whether scaling the latent code corresponds to changing $\log p_0(\boldsymbol{x}_0)$. In particular, we showed that the necessary SA condition Eq. 11 does not always hold. Furthermore, the prior guidance does *not* allow choosing the desired sample log-density.

We now present an approach for sampling $\boldsymbol{x}_0$ with explicit control of $\log p_0(\boldsymbol{x}_0)$. Suppose that we require an instantaneous density changes over time,

$$\text{constraint:} \quad \frac{d \log p_t(\boldsymbol{x}_t)}{dt} = b_t(\boldsymbol{x}_t) \quad \in \mathbb{R} \quad (14)$$

for a predetermined $b_t$. To achieve this, we choose a new ODE $d\boldsymbol{x}_t = \tilde{\boldsymbol{u}}_t dt$, such that its density change from Eq. 4 satisfies

$$b_t(\boldsymbol{x}_t) = \nabla \log p_t(\boldsymbol{x}_t)^T \left( \tilde{\boldsymbol{u}}_t(\boldsymbol{x}_t) - \boldsymbol{u}_t(\boldsymbol{x}_t) \right) - \text{div}\, \boldsymbol{u}_t(\boldsymbol{x}_t). \quad (15)$$

Whenever $\nabla \log p_t(\boldsymbol{x}_t) \neq \mathbf{0}$, Eq. 15 has multiple solutions of $\tilde{\boldsymbol{u}}$ for any $b_t$. We choose $\tilde{\boldsymbol{u}}$ that is closest to $\boldsymbol{u}$, which uniquely gives (See Appendix D)

$$\tilde{\boldsymbol{u}}_t(\boldsymbol{x}) = \boldsymbol{u}_t(\boldsymbol{x}) + \underbrace{\frac{\text{div}\, \boldsymbol{u}_t(\boldsymbol{x}) + b_t(\boldsymbol{x})}{\|\nabla \log p_t(\boldsymbol{x})\|^2}}_{\text{score bias } s_b(\boldsymbol{x})} \nabla \log p_t(\boldsymbol{x}). \quad (16)$$

> **Density guidance:** Eq. 16 steers the sample away from the original trajectory towards desired likelihood.

When $b_t = -\text{div}\, \boldsymbol{u}_t$, we reduce to the canonical sampler $\tilde{\boldsymbol{u}}_t = \boldsymbol{u}_t$. Since using Eq. 16 requires knowing the score function, we assume from this point on that $\boldsymbol{u}_t$ is given by the PF-ODE of a diffusion Eq. 6, which is transformed by

---

**Algorithm 1** Score Alignment verification

1: **input:** Flow $\boldsymbol{u}_t$, latent $\boldsymbol{x}_T \in \mathbb{R}^D$, step size $dt > 0$
2: **initialize** $\boldsymbol{v}_T = \nabla \log p_T(\boldsymbol{x}_T)$, $t = T$, $\omega_T = \|\boldsymbol{v}_T\|^2$
3: **while** $t > 0$ **do**
4: $\quad d\boldsymbol{x} \leftarrow \boldsymbol{u}_t(\boldsymbol{x}_t)$
5: $\quad d\boldsymbol{v} \leftarrow \text{JVP}(\boldsymbol{u}_t, \boldsymbol{x}_t, \boldsymbol{v}_t)$
6: $\quad \boldsymbol{\varepsilon} \leftarrow \text{Uniform}\{-1, 1\}^D$ $\quad$ *Rademacher variables*
7: $\quad d\omega \leftarrow -\boldsymbol{\varepsilon}^T \text{JVP}(d\boldsymbol{v}, \boldsymbol{x}_t, \boldsymbol{\varepsilon})$ $\quad$ *Hutchinson's trick*
8: $\quad \boldsymbol{x}_t \leftarrow \boldsymbol{x}_t - dt \cdot d\boldsymbol{x}$
9: $\quad \boldsymbol{v}_t \leftarrow \boldsymbol{v}_t - dt \cdot d\boldsymbol{v}$
10: $\quad \omega_t \leftarrow \omega_t - dt \cdot d\omega$
11: $\quad t \leftarrow t - dt$
12: **end while**
13: **if** $\nabla \log p_0(\boldsymbol{x}_0)$ known **then**
14: $\quad$ **return** $\nabla \log p_0(\boldsymbol{x}_0)^T \boldsymbol{v}_0$
15: **else**
16: $\quad$ **return** $\omega_0$
17: **end if**

---

*Figure 6.* **Score Alignment Verification.** When the score $\nabla \log p_0$ is known, Eq. 12 applies, and the highlighted steps (corresponding to Eq. 13) can be omitted. We provide JAX implementation in Listing 1.

Eq. 16 into

$$\tilde{\boldsymbol{u}}_t(\boldsymbol{x}_t) = f(t)\boldsymbol{x}_t + \left( s_b(\boldsymbol{x}_t) - \frac{1}{2} g^2(t) \right) \nabla \log p_t(\boldsymbol{x}_t), \quad (17)$$

which can readily be used for sampling for any $b$.

The question is: How to choose $b_t$? Notably, we cannot simply push $b_t$ to be arbitrarily high or low, since it will fall off the diffusion manifold, leading to nonsense decodings.

## 4.1. Explicit quantile matching

Suppose that we want $\boldsymbol{x}_0$ to have a pre-defined value of log-density $\log p_0(\boldsymbol{x}_0) = c \in \mathbb{R}$, which is equivalent to

$$\int_0^T b_t(\boldsymbol{x}_t) dt = \log p_T(\boldsymbol{x}_T) - c. \quad (18)$$

If this holds for $b$, the Eq. 16 will generate a sample $\boldsymbol{x}_0$ with the log-density $c$. However, not all choices of $b$ are equally good. In practice, $\boldsymbol{u}$ and $\nabla \log p_t$ are approximated with neural networks, and their predictions are only accurate when $\boldsymbol{x}_t$ is in the typical region of $p_t$ (Nalisnick et al., 2019).

Suppose that the target value $c$ is the $q$'th quantile of $\log p_0$, where $q \in [0, 1]$. A simple strategy is to choose $b_t$ such that the sample $\boldsymbol{x}_t$ remains on the same quantile $q$ over all times

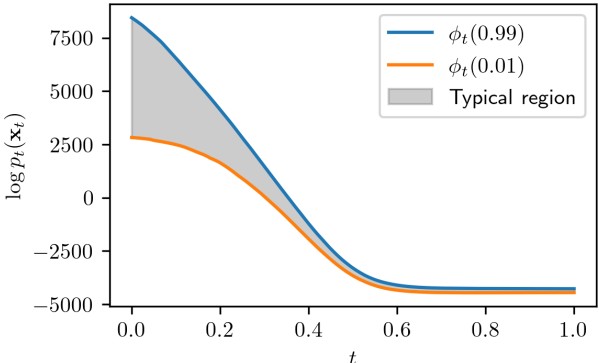

Figure 7. Quantiles and typical values of $\log p_t(\boldsymbol{x}_t)$ for a diffusion model trained on CIFAR10.

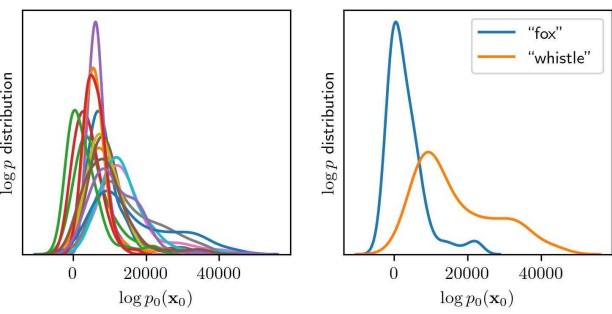

Figure 8. **Different classes have different likelihoods.** Left: Distributions of log-likelihoods of 16 randomly selected classes from ImageNet. Right: Distributions of log-likelihoods for "fox" and "whistle" differ significantly.

$t$ and $p_t$. Let $\phi_t(q)$ denote the $q$-th quantile of $\log p_t$. Then

$$\log p_t(\boldsymbol{x}_t) = \log p_T(\boldsymbol{x}_T) - \int_t^T b_\tau(\boldsymbol{x}_\tau)d\tau = \phi_t(q), \quad (19)$$

which is satisfied for $b_t(\boldsymbol{x}) := \frac{d}{dt}\phi_t(q)$. The quantile function $\phi_t(q)$ can be estimated by sampling $K$ independent samples $\boldsymbol{x}_T \sim p_T$, estimating $\log p_t(\boldsymbol{x}_t)$ with Eq. 3 and finding empirical quantiles for target values of $q$. We visualize $\phi_t(0.99)$ and $\phi_t(0.01)$ in Fig. 7 estimated for a Variance-Preserving (VP) SDE diffusion model with linear log-SNR schedule trained on CIFAR10. We experimentally verify the accuracy of explicit quantile matching in Appendix E.

### 4.2. Implicit quantile matching

A considerable drawback of explicit quantile matching is the need to estimate $\phi_t$. This becomes especially problematic for conditional generation, where the distribution of $\log p_t$ can differ significantly for different classes (Fig. 8). For applications such as text-to-image, this would require estimating the distribution of $\log p_t$ for every possible text prompt, which is not feasible.

The Eq. 15 gives a recipe for altering the flow based on how

we want $\log p_t(\boldsymbol{x}_t)$ to evolve. However, the challenge is to determine what are the *reasonable* values of $b_t$ so that $\log p_t$ does not deviate from what is typical. We tackle this problem by analyzing the stochastic view of CNFs. Specifically, for $\boldsymbol{x}_t \sim p_t$, Eq. 7 says that when evaluating $\boldsymbol{x}_{t-dt}$, we can add random noise and stay within the typical region of $p_{t-dt}$ as long as we correct for it by subtracting the score from the drift. Furthermore, Eq. 8 says how $\log p_t$ changes under this stochastic evolution.

Concretely, the *average* change in log-density, when adding noise of strength $\varphi(t)$ is given by

$$\mathbb{E}\big[d\log p_t(\boldsymbol{x}_t)\big] = -\left(\text{div }\boldsymbol{u}_t(\boldsymbol{x}_t) + \frac{1}{2}\varphi^2(t)h(\boldsymbol{x}_t)\right)dt, \quad (20)$$

where $h(\boldsymbol{x}) = \Delta\log p_t(\boldsymbol{x}) + \|\nabla\log p_t(\boldsymbol{x})\|^2 \in \mathbb{R}$. Interestingly, in Appendix F, we empirically (and theoretically in simplified cases) show that in diffusion models $\frac{\sigma_t^2 h(\boldsymbol{x}_t)}{\sqrt{2D}}$ approximately follows $\mathcal{N}(0,1)$ when $\boldsymbol{x}_t \sim p_t$ and dimension $D$ is high. A reasonable choice is then

$$b_t(\boldsymbol{x}) = -\text{div }\boldsymbol{u}_t(\boldsymbol{x}) - \frac{1}{2}\varphi^2(t)\frac{\sqrt{2D}}{\sigma_t^2}\Phi^{-1}(q), \quad (21)$$

where $\Phi$ is the cumulative distribution function of $\mathcal{N}(0,1)$ and $q$ is the desired quantile. We found that choosing $\varphi$ to match the diffusion strength in Eq. 5, i.e. $\varphi \equiv g$ works well in practice. Thus, in our experiments we use

$$b_t^q(\boldsymbol{x}) = -\text{div }\boldsymbol{u}_t(\boldsymbol{x}) - \frac{1}{2}g^2(t)\frac{\sqrt{2D}}{\sigma_t^2}\Phi^{-1}(q). \quad (22)$$

After plugging this definition of $b$ to Eq. 16, we get

$$\boldsymbol{u}_t^{\text{DG-ODE}}(\boldsymbol{x}_t) = f(t)\boldsymbol{x}_t - \frac{1}{2}g^2(t)\eta_t(\boldsymbol{x}_t)\nabla\log p_t(\boldsymbol{x}_t), \quad (23)$$

which is equivalent to simply rescaling the score by

$$\eta_t(\boldsymbol{x}) = 1 + \frac{\sqrt{2D}\Phi^{-1}(q)}{\|\sigma_t\nabla\log p_t(\boldsymbol{x})\|^2}. \quad (24)$$

We call Eq. 23 *Density-Guided Sampling* (DGS). Importantly, DGS comes at no extra cost since the score is evaluated at each sampling step anyway. Note that, as shown in Fig. 4, the correlation of $\log p_t(\boldsymbol{x}_t)$ with image detail is the strongest for $t^* \approx \log \text{SNR}^{-1}(1)$ and thus in our experiments we only use guidance in the $[T, t^*]$ interval. In Fig. 9 we show samples generated with DGS with different values of $q$. Interestingly, the samples are perceptually very similar to those from Prior Guidance (Appendix M). See Appendix J for more samples and quantitative results.

**Conditional generation** Whenever a conditional score function $\nabla\log p_t(\boldsymbol{x}|\text{cond})$ is available, where cond can be any condition (class, text, etc.), one need only replace the score function with the conditional one in Eq. 23.

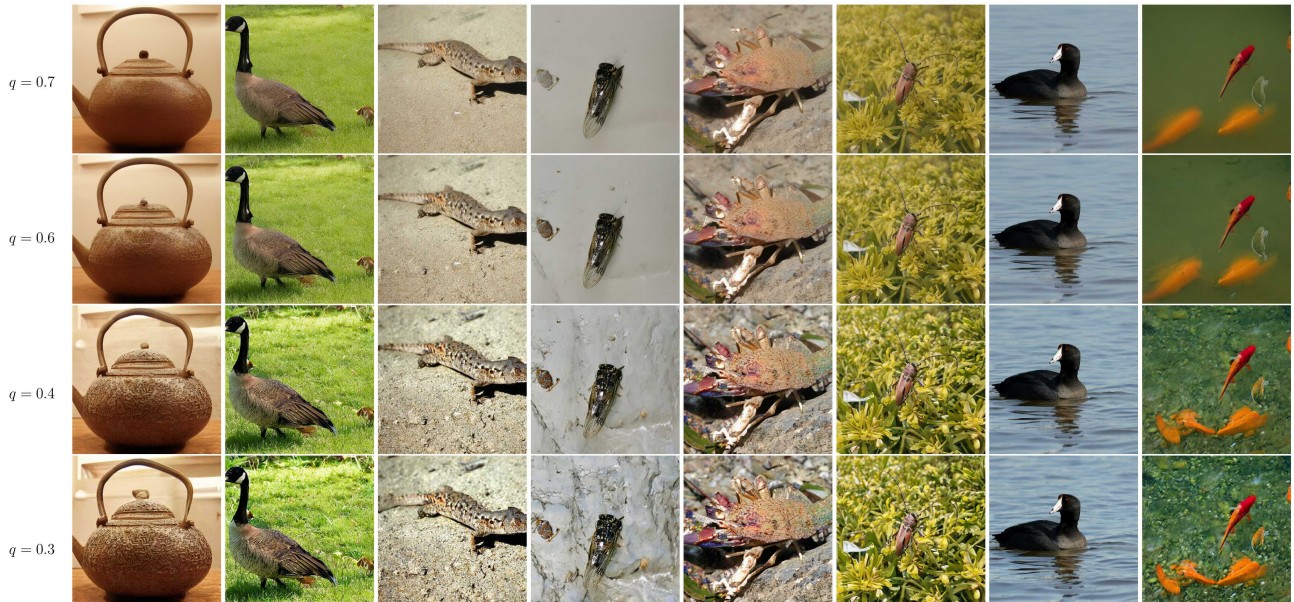

*Figure 9.* **Density Guidance controls the amount of detail.** Samples generated with Eq. 23 using the EDM2 model (Karras et al., 2024b).

## 5. Stochastic density guidance

In previous sections we discussed two methods for controlling $\log p_0(\boldsymbol{x}_0)$ during ODE sampling of form Eq. 1. However, it has been reported that adding stochasticity during sampling can improve sample quality (Song et al., 2021b; Karras et al., 2022). Neither of the previously discussed methods supports stochastic sampling. Recently Karczewski et al. (2025) lifted the first roadblock towards this by showing how to *evaluate* $\log p_0(\boldsymbol{x}_0)$ for an SDE. We now ask: Is it possible to also *control* $\log p_0(\boldsymbol{x}_0)$ during stochastic sampling?

Recall a stochastic CNF sampler with noise strength $\varphi$:

$$d\boldsymbol{x}_t = \Big(\boldsymbol{u}_t(\boldsymbol{x}_t) - \frac{1}{2}\varphi^2(t)\nabla \log p_t(\boldsymbol{x}_t)\Big)dt + \varphi(t)d\overline{\mathbf{W}}_t. \quad (7)$$

In Appendix G, we show that, similarly to density guidance Eq. 23, it can be altered to enforce the desired evolution of log-density over time. Specifically, suppose that $\boldsymbol{u}_t = \boldsymbol{u}_t^{\text{PF-ODE}}$ and that we require $\frac{d\log p_t(\boldsymbol{x}_t)}{dt} = b_t(\boldsymbol{x}_t)$ for $b$ defined in Eq. 22. Then, the stochastic process

$$d\boldsymbol{x}_t = \boldsymbol{u}_t^{\text{DG-SDE}}(\boldsymbol{x}_t)dt + \varphi(t)P_t(\boldsymbol{x}_t)d\overline{\mathbf{W}}_t \quad (25)$$

*approximately* satisfies $\frac{d\log p_t(\boldsymbol{x}_t)}{dt} = b_t(\boldsymbol{x}_t)$, where

$$\boldsymbol{u}_t^{\text{DG-SDE}}(\boldsymbol{x}) = \boldsymbol{u}_t^{\text{DG-ODE}}(\boldsymbol{x}) \quad (26)$$

$$\underbrace{+ \frac{1}{2}\varphi^2(t)\frac{\Delta \log p_t(\boldsymbol{x})}{\|\nabla \log p_t(\boldsymbol{x})\|^2}\nabla \log p_t(\boldsymbol{x})}_{\text{correction for added stochasticity}} \quad (27)$$

and

$$P_t(\boldsymbol{x}) = \boldsymbol{I}_D - \Big(\frac{\nabla \log p_t(\boldsymbol{x})}{\|\nabla \log p_t(\boldsymbol{x})\|}\Big)\Big(\frac{\nabla \log p_t(\boldsymbol{x})}{\|\nabla \log p_t(\boldsymbol{x})\|}\Big)^T \quad (28)$$

is the "score-orthogonal" projection, which ensures that the $\log p_t(\boldsymbol{x}_t)$ changes deterministically even though $\boldsymbol{x}_t$ is stochastic. In Appendix G we provide the formula Eq. 120 of the SDE drift that *exactly* achieves $d\log p_t(\boldsymbol{x}_t) = b_t(\boldsymbol{x}_t)dt$ for any choice of $b_t$ and $\boldsymbol{u}_t$, which we omit here for presentation clarity. In Appendix H we experimentally demonstrate that we can obtain exact likelihoods even for stochastic sampling, provided the number of sampling steps is large enough.

The Eq. 25 allows increasing variation in the samples by injecting noise to DGS Eq. 23 whilst maintaining the desired evolution of the log-density. Furthermore, since it is known that diffusion models first generate high-level features and then the details (Ho et al., 2020; Deja et al., 2022; Wang & Vastola, 2023), DGS can be combined with stochasticity by introducing noise at specific stages of the generation process, allowing for controlled variation in either high- or low-level features while preserving the desired level of detail. We demonstrate this approach in Fig. 10 and provide more samples in Appendix K.

## 6. Related Work

Sehwag et al. (2022) proposed a method for generating samples from low-density regions of diffusion models. However, due to the intractability of likelihood in diffusion models, their approach relies on approximations. Subsequent work

Deterministic guidance  Adding noise early (high-level variation)  Adding noise late (low-level variation)

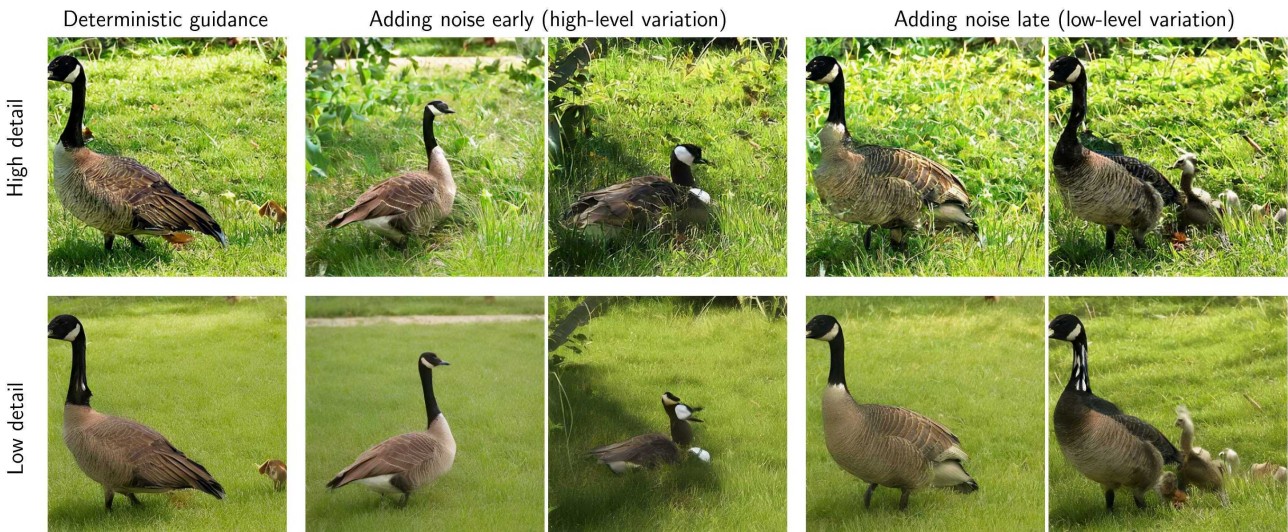

*Figure 10.* **Stochastic density guidance increases variation in generated samples whilst maintaining the desired level of detail.** Samples generated with Eq. 25. Adding stochasticity early in the sampling process changes high-level features, whereas adding noise later, only affects lower-level detail.

by Karczewski et al. (2025) demonstrated that likelihood is, in fact, tractable even in stochastic diffusion models, challenging the need for such approximations. Building on this, our proposed methods provide explicit likelihood control for both deterministic sampling—via Prior Guidance (section 3) and Density Guidance (section 4)—and stochastic sampling through Stochastic Density Guidance (section 5).

Song et al. (2021b) observed that scaling the latent code alters the amount of detail in deterministically generated images. While this phenomenon has been widely acknowledged, we provide a rigorous analysis (section 3) and prove that it is a direct consequence of Score Alignment Eq. 11, which guarantees that scaling leads to a monotonic change in the likelihood of the generated image, $\log p_0(\boldsymbol{x}_0)$. Furthermore, we introduce tractable numerical tools (Fig. 6) that can verify whether any given CNF model (not necessarily score-based) exhibits this behavior.

Karras et al. (2024a) proposed auto-guidance as a method for improving sample quality by targeting high-density regions. However, Karczewski et al. (2025) found that the highest-density regions in diffusion models contain cartoon-like or blurry images, which raises concerns about the effectiveness of purely maximizing likelihood. In contrast, we introduce multiple cost-free methods for explicitly controlling the likelihood of generated samples. Additionally, while Karras et al. (2024a) observed that scaling the score function leads to oversimplified images, we demonstrate that DGS Eq. 23 enables effective control over image detail—both increasing and decreasing it—when the scaling is adapted both temporally and spatially Eq. 24.

Yu et al. (2023) introduced Riemannian Langevin Dynamics, an SDE with a non-diagonal diffusion matrix, similar in structure to our Stochastic Density Guidance (section 5). However, a key distinction is that our diffusion matrix is a projection onto the orthogonal complement of the subspace spanned by the score function. As a result, it is not positive definite and cannot serve as a Riemannian metric tensor, making our approach fundamentally different in its mathematical formulation and behavior.

Recently, Kamkari et al. (2024) proposed a method for measuring local intrinsic dimension, which, in the case of images, corresponds to the amount of detail present. However, we show that negative $\log p$ is a more effective measure of image detail and provide empirical comparisons in Fig. 4. Moreover, while Kamkari et al. (2024) focus on measuring image detail, our methods enable direct manipulation of it, allowing for finer control over generative model outputs.

# 7. Conclusion

In this paper, we introduced methods for controlling sample density in flow models, enabling manipulation of image detail through likelihood-guided sampling. We provided a theoretical explanation of latent code scaling by introducing score alignment, a condition that can be tractably checked for any CNF model. Building on this, we derived Density Guidance, a principled modification of the generative ODE that allows for exact log-density control during sampling. Finally, we extended this approach to stochastic sampling, demonstrating that it retains precise detail control while allowing controlled variation in image structure and detail.

Our findings deepen the understanding of likelihood in flow models and provide practical tools for better sample control.

## Impact Statement

This paper presents work that advances the understanding and controllability of sample density in diffusion-based generative models. By introducing techniques for precise log-density control, our work contributes to improved interpretability and fine-grained control over image generation. While these advancements could enhance applications in creative and scientific domains, they also raise considerations around synthetic media generation and potential misuse. However, our contributions primarily aim at improving theoretical understanding and control in generative modeling, without introducing new ethical risks beyond those already associated with generative AI.

## Acknowledgements

This work was supported by the Finnish Center for Artificial Intelligence (FCAI) under Flagship R5 (award 15011052). RK thanks Paulina Karczewska for her help with preparing figures. VG acknowledges the support from Saab-WASP (grant 411025), Academy of Finland (grant 342077), and the Jane and Aatos Erkko Foundation (grant 7001703).

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

# A. Auxiliary results

## A.1. Constrained optimization

In multiple sections, we will be solving constrained optimization problems, which can be written in the following way. Suppose $\boldsymbol{v} \in \mathbb{R}^D$, $\boldsymbol{v} \neq \boldsymbol{0}$, any $\boldsymbol{y} \in \mathbb{R}^D$ and $a \in \mathbb{R}$. The problem we will encounter is

$$
\min_{\boldsymbol{x} \in \mathbb{R}^D} \frac{1}{2} \|\boldsymbol{x} - \boldsymbol{y}\|^2
$$
$$
\text{s.t.} \quad \boldsymbol{x}^T \boldsymbol{v} = a.
$$
(29)

We solve this by introducing the Lagrangian $\mathcal{L}(\boldsymbol{x}, \lambda) = \frac{1}{2} \|\boldsymbol{x} - \boldsymbol{y}\|^2 + \lambda(\boldsymbol{x}^T \boldsymbol{v} - a)$ for $\lambda \in \mathbb{R}$. By setting $\frac{\partial \mathcal{L}}{\partial \boldsymbol{x}} = \boldsymbol{0}$, we get

$$
\boldsymbol{x} - \boldsymbol{y} + \lambda \boldsymbol{v} = 0 \Rightarrow \boldsymbol{x} = \boldsymbol{y} - \lambda \boldsymbol{v}.
$$
(30)

To find $\lambda$ we substitute for $\boldsymbol{x}$ in the constraint and find

$$
\boldsymbol{y}^T \boldsymbol{v} - \lambda \|\boldsymbol{v}\|^2 = a \Rightarrow \lambda = \frac{\boldsymbol{y}^T \boldsymbol{v} - a}{\|\boldsymbol{v}\|^2}.
$$
(31)

Combining the two, we get that the solution is given by

$$
\boldsymbol{x} = \boldsymbol{y} + \frac{a - \boldsymbol{v}^T \boldsymbol{y}}{\|\boldsymbol{v}\|^2} \boldsymbol{v}.
$$
(32)

## A.2. Divergence-gradient identity

We will make use of an identity connecting the gradient of the divergence with the divergence of a Jacobian vector product.

**Lemma A.1.** *Let $f : \mathbb{R}^D \to \mathbb{R}^D$ with continuous 2-nd order derivatives and $\boldsymbol{v} \in \mathbb{R}$. Define $g(\boldsymbol{x}) := \operatorname{div} f(\boldsymbol{x}) = \sum_{i=1}^{D} \frac{\partial f^i}{\partial x_i}(\boldsymbol{x})$ and $G(\boldsymbol{x}) := \frac{\partial f}{\partial \boldsymbol{x}}(\boldsymbol{x})\boldsymbol{v}$. Then $g : \mathbb{R}^D \to \mathbb{R}$ is a scalar function and $G : \mathbb{R}^D \to \mathbb{R}^D$ is a vector function satisfying*

$$
\nabla g(\boldsymbol{x})^T \boldsymbol{v} = \operatorname{div} G(\boldsymbol{x}).
$$
(33)

*Equivalently, we write it as*

$$
(\nabla \operatorname{div} f(\boldsymbol{x}))^T \boldsymbol{v} = \operatorname{div} \left( \frac{\partial f}{\partial \boldsymbol{x}}(\boldsymbol{x})\boldsymbol{v} \right)
$$
(34)

*Proof.*

$$
\nabla g(\boldsymbol{x})^T \boldsymbol{v} = \sum_{j=1}^{D} \frac{\partial g}{\partial x_j}(\boldsymbol{x}) v_j = \sum_{j=1}^{D} \frac{\partial}{\partial x_j} \left( \sum_{i=1}^{D} \frac{\partial f^i}{\partial x_i}(\boldsymbol{x}) \right) v_j = \sum_{i=1}^{D} \frac{\partial}{\partial x_i} \left( \sum_{j=1}^{D} \frac{\partial f^i}{\partial x_j} v_j \right) = \sum_{i=1}^{D} \frac{\partial}{\partial x_i} \left( \frac{\partial f}{\partial \boldsymbol{x}}(\boldsymbol{x})\boldsymbol{v} \right)_i
$$
$$
= \sum_{i=1}^{D} \frac{\partial}{\partial x_i} G^i(\boldsymbol{x}) = \operatorname{div} G(\boldsymbol{x}).
$$

$\square$

## A.3. Optimality of projection

In Appendix G we will be interested in finding a linear operator $\boldsymbol{A} \in \mathbb{R}^{D \times D}$ satisfying $\boldsymbol{A}\boldsymbol{v} = \boldsymbol{0}$ for some $\boldsymbol{v}$, so that the distance between $\boldsymbol{A}$ and the identity $\boldsymbol{I}_D$ is minimal. The following lemma provides a solution.

**Lemma A.2.** *Let $\boldsymbol{0} \neq \boldsymbol{v} \in \mathbb{R}^D$. The solution of*

$$
\min_{\boldsymbol{A} \in \mathbb{R}^{D \times D}} \|\boldsymbol{A} - \boldsymbol{I}_D\|
$$
$$
\text{s.t. } \boldsymbol{A}\boldsymbol{v} = \boldsymbol{0},
$$
(35)

*where $\| \cdot \|$ can be either the spectral or Frobenius norm, is given by the projection matrix*

$$
\boldsymbol{A}^{\text{OPT}} = \boldsymbol{P} = \boldsymbol{I}_D - \left( \frac{\boldsymbol{v}}{\|\boldsymbol{v}\|} \right) \left( \frac{\boldsymbol{v}}{\|\boldsymbol{v}\|} \right)^T.
$$
(36)

*Proof.* First note that for any $\boldsymbol{A} \in \mathbb{R}^{D \times D}$ satisfying $\boldsymbol{A}\boldsymbol{v} = \boldsymbol{0}$, we have

$$\|\boldsymbol{A} - \boldsymbol{I}_D\|_F \geq \|\boldsymbol{A} - \boldsymbol{I}_D\|_2 = \max_{\boldsymbol{w} \neq 0} \left| \frac{\boldsymbol{w}^T(\boldsymbol{A} - \boldsymbol{I}_D)\boldsymbol{w}}{\|\boldsymbol{w}\|^2} \right| \geq \left| \frac{\boldsymbol{v}^T(\boldsymbol{A} - \boldsymbol{I}_D)\boldsymbol{v}}{\|\boldsymbol{v}\|^2} \right| = \left| \frac{\boldsymbol{v}^T(0 - \boldsymbol{v})}{\|\boldsymbol{v}\|^2} \right| = \frac{\boldsymbol{v}^T\boldsymbol{v}}{\|\boldsymbol{v}\|^2} = 1. \quad (37)$$

On the other hand $\boldsymbol{P} - \boldsymbol{I}_D = \left( \frac{\boldsymbol{v}}{\|\boldsymbol{v}\|} \right) \left( \frac{\boldsymbol{v}}{\|\boldsymbol{v}\|} \right)^T$, which has only a single non-zero eigenvalue $\lambda = 1$ and thus

$$\|\boldsymbol{P} - \boldsymbol{I}_D\|_F = \|\boldsymbol{P} - \boldsymbol{I}_D\|_2 = 1 \quad (38)$$

and

$$\boldsymbol{P}\boldsymbol{v} = \boldsymbol{v} - \frac{\boldsymbol{v}^T\boldsymbol{v}}{\|\boldsymbol{v}\|^2}\boldsymbol{v} = \boldsymbol{0}. \quad (39)$$

Therefore $\boldsymbol{A} = \boldsymbol{P}$ satisfies $\boldsymbol{A}\boldsymbol{v} = \boldsymbol{0}$ and minimizes both $\|\boldsymbol{A} - \boldsymbol{I}_D\|_2$ and $\|\boldsymbol{A} - \boldsymbol{I}_D\|_F$. $\qquad \square$

# B. Derivation of CNF density evolutions

We reproduce the continuous-time normaling flow (CNF) density evolution of Chen et al. (2018),

$$\frac{d \log p_t(\boldsymbol{x}_t)}{dt} = -\operatorname{div} \boldsymbol{u}_t(\boldsymbol{x}_t), \quad (40)$$

and the generalised CNF density evolution of Karczewski et al. (2025),

$$\frac{d \log p_t(\boldsymbol{x}_t)}{dt} = -\operatorname{div} \boldsymbol{u}_t(\boldsymbol{x}_t) + \nabla \log p_t(\boldsymbol{x}_t)^T (\tilde{\boldsymbol{u}}_t(\boldsymbol{x}_t) - \boldsymbol{u}_t(\boldsymbol{x}_t)), \quad (41)$$

with a unified derivation.

We assume a time-dependent particle $\boldsymbol{x}_t \in \mathbb{R}^D$ evolving through continuous time $t \in \mathbb{R}$ governed by an ordinary differential equation (ODE)

$$\frac{d\boldsymbol{x}_t}{dt} = \boldsymbol{u}_t(\boldsymbol{x}_t), \quad (42)$$

where $\boldsymbol{u}_t(\boldsymbol{x}) : \mathbb{R} \times \mathbb{R}^D \mapsto \mathbb{R}^D$ is a time-dependent vector field that maps any state vector $\boldsymbol{x}$ to its time derivative vector $\boldsymbol{u}_t(\boldsymbol{x}_t)$.

**CNF density evolution** We are interested in the time evolution of the spatiotemporal log-likelihood $\log p_t(\boldsymbol{x}_t)$ for particles evolving under the ODE. We write the log density total derivative wrt time by using the chain rule

$$\frac{d \log p_t(\boldsymbol{x}_t)}{dt} = \frac{1}{p_t(\boldsymbol{x}_t)} \frac{dp_t(\boldsymbol{x}_t)}{dt} \quad (43)$$

$$= \frac{1}{p_t(\boldsymbol{x}_t)} \left( \frac{\partial p_t(\boldsymbol{x}_t)}{\partial t} + \frac{\partial p_t(\boldsymbol{x}_t)}{\partial \boldsymbol{x}} \cdot \frac{d\boldsymbol{x}_t}{dt} \right), \quad (44)$$

which describes the density evolution of a particle moving under a flow. We assume that the ODE is a continuous-time normalizing flow, where the density is conserved over time. This is described by the continuity equation (Finlay et al., 2020; Xu et al., 2024)

$$\frac{\partial p_t(\boldsymbol{x}_t)}{\partial t} + \nabla \cdot \left( p_t(\boldsymbol{x}_t)\boldsymbol{u}_t(\boldsymbol{x}_t) \right) = 0, \quad (45)$$

which describes the change in particle density as a result of a vector field $\boldsymbol{u}_t$ transporting the particles, at location $\boldsymbol{x}$. By substitution we obtain

$$\frac{d\log p_t(\boldsymbol{x}_t)}{dt} = \frac{1}{p_t(\boldsymbol{x}_t)}\Big( -\nabla \cdot (p_t(\boldsymbol{x}_t)\boldsymbol{u}_t(\boldsymbol{x}_t)) + \nabla p_t(\boldsymbol{x}_t) \cdot \boldsymbol{u}_t(\boldsymbol{x}_t) \Big) \tag{46}$$

$$= \frac{1}{p_t(\boldsymbol{x}_t)}\Big( -p_t(\boldsymbol{x}_t)\nabla \cdot \boldsymbol{u}_t(\boldsymbol{x}_t) - \nabla p_t(\boldsymbol{x}_t) \cdot \boldsymbol{u}_t(\boldsymbol{x}_t) + \nabla p_t(\boldsymbol{x}_t) \cdot \boldsymbol{u}_t(\boldsymbol{x}_t) \Big) \tag{47}$$

$$= -\frac{1}{p_t(\boldsymbol{x}_t)}p_t(\boldsymbol{x}_t)\nabla \cdot \boldsymbol{u}_t(\boldsymbol{x}_t) \tag{48}$$

$$= -\nabla \cdot \boldsymbol{u}_t(\boldsymbol{x}_t) \tag{49}$$

$$= -\operatorname{div}\boldsymbol{u}_t(\boldsymbol{x}_t). \tag{50}$$

**Generalised CNF density evolution**  Next, we derive the evolution of log-density $\log p_t$ of a particle that is moving in some non-canonical direction, ie. $\dot{\boldsymbol{x}}_t = \tilde{\boldsymbol{u}}_t(\boldsymbol{x}_t) \neq \boldsymbol{u}_t(\boldsymbol{x}_t)$. Notably, the continuity equation remains with the $\boldsymbol{u}_t$ as we are describing the particle density in the marginal induced by the original transport $\boldsymbol{u}_t$. We obtain

$$\frac{d\log p_t(\boldsymbol{x}_t)}{dt} = \frac{1}{p_t(\boldsymbol{x}_t)}\Big( -\nabla \cdot \big(p_t(\boldsymbol{x}_t)\boldsymbol{u}_t(\boldsymbol{x}_t)\big) + \nabla p_t(\boldsymbol{x}_t) \cdot \tilde{\boldsymbol{u}}_t(\boldsymbol{x}_t) \Big) \tag{51}$$

$$= \frac{1}{p_t(\boldsymbol{x}_t)}\Big( -p_t(\boldsymbol{x}_t)\nabla \cdot \boldsymbol{u}_t(\boldsymbol{x}_t) - \nabla p_t(\boldsymbol{x}_t) \cdot \boldsymbol{u}_t(\boldsymbol{x}_t) + \nabla p_t(\boldsymbol{x}_t) \cdot \tilde{\boldsymbol{u}}_t(\boldsymbol{x}_t) \Big) \tag{52}$$

$$= \frac{1}{p_t(\boldsymbol{x}_t)}\Big( -p_t(\boldsymbol{x}_t)\nabla \cdot \boldsymbol{u}_t(\boldsymbol{x}_t) + \nabla p_t(\boldsymbol{x}_t) \cdot \big( \tilde{\boldsymbol{u}}_t(\boldsymbol{x}_t) - \boldsymbol{u}_t(\boldsymbol{x}_t) \big) \Big) \tag{53}$$

$$= -\nabla \cdot \boldsymbol{u}_t(\boldsymbol{x}_t) + \frac{1}{p_t(\boldsymbol{x}_t)}\nabla p_t(\boldsymbol{x}_t) \cdot \big( \tilde{\boldsymbol{u}}_t(\boldsymbol{x}_t) - \boldsymbol{u}_t(\boldsymbol{x}_t) \big) \tag{54}$$

$$= -\operatorname{div}\boldsymbol{u}_t(\boldsymbol{x}_t) + \nabla \log p_t(\boldsymbol{x}_t) \cdot \big( \tilde{\boldsymbol{u}}_t(\boldsymbol{x}_t) - \boldsymbol{u}_t(\boldsymbol{x}_t) \big), \tag{55}$$

which is the generalised instantaneous change of density.

## C. Derivation of Score Alignment

In this section, we prove the Score Alignment condition, a necessary and sufficient condition for prior guidance to be effective in controlling $\log p_0$. Formally, assume a latent curve $c : [0,1] \to \mathbb{R}^D$ following the score at $t = T$, $c'(s) = \nabla \log p_T(\boldsymbol{x}_0(c(s)))$. In a Gaussian prior $p_T$ the curve becomes a line of scaled latents $\boldsymbol{x}_T$. The $\log p_0$ is monotonic on the decoded curve when

$$\frac{d}{ds}\log p_0(\boldsymbol{x}_0(c(s))) \geq 0, \qquad \forall s \in (0,1). \tag{56}$$

The chain rule gives the derivative

$$\frac{d}{ds}\log p_0(\boldsymbol{x}_0(c(s))) = \nabla \log p_0(\boldsymbol{x}_0(c(s)))^T \frac{d}{ds}\boldsymbol{x}_0(c(s)) \tag{57}$$

$$= \nabla \log p_0(\boldsymbol{x}_0(c(s)))^T \frac{\partial \boldsymbol{x}_0}{\partial \boldsymbol{x}_T}(c(s))c'(s) \tag{58}$$

$$= \nabla \log p_0(\boldsymbol{x}_0(c(s)))^T \frac{\partial \boldsymbol{x}_0}{\partial \boldsymbol{x}_T}(c(s))\nabla \log p_T(c(s)) \tag{59}$$

Therefore, for Eq. 56 to hold for a curve passing through some arbitrary $\boldsymbol{x}_T \in \mathbb{R}^D$ at some point $s \in (0,1)$ it must hold

$$\nabla \log p_0(\boldsymbol{x}_0(\boldsymbol{x}_T))^T \frac{\partial \boldsymbol{x}_0}{\partial \boldsymbol{x}_T}(\boldsymbol{x}_T)\nabla \log p_T(\boldsymbol{x}_T) \geq 0, \tag{60}$$

where in Eq. 11 we omit the $\boldsymbol{x}_T$ in the parentheses for brevity.

## C.1. Score alignment time evolution

In this subsection, we derive Eq. 13, i.e. how the SA condition can be checked without knowing $\nabla \log p_t$ for $t < T$. Let $c$ be a latent curve following the score and passing through $\boldsymbol{x}_T$, i.e. $c : (-\varepsilon, \varepsilon) \to \mathbb{R}^D$, $c'(s) = \nabla \log p_T(c(s))$ and $c(0) = \boldsymbol{x}_T$. Define

$$\psi(t, s) := \log p_t(\boldsymbol{x}_t(c(s))) \text{ for } t \in [0, T], \ s \in (-\varepsilon, \varepsilon). \tag{61}$$

The SA condition at $\boldsymbol{x}_T$ (Eq. 60) is given by

$$\frac{\partial \psi}{\partial s}(0, 0) = \nabla \log p_0(\boldsymbol{x}_0(\boldsymbol{x}_T))^T \frac{\partial \boldsymbol{x}_0}{\partial \boldsymbol{x}_T}(\boldsymbol{x}_T) \nabla \log p_T(\boldsymbol{x}_T). \tag{62}$$

Note that

$$\frac{\partial \psi}{\partial s}(T, 0) = \frac{d}{ds} \log p_T(c(s)) \Big|_{s=0} = \nabla \log p_T(c(s))^T c'(s) \Big|_{s=0} = \|\nabla \log p_T(\boldsymbol{x}_T)\|^2. \tag{63}$$

Therefore Eq. 62 can be equivalently written as

$$\frac{\partial \psi}{\partial s}(0, 0) = \frac{\partial \psi}{\partial s}(T, 0) + \left( \frac{\partial \psi}{\partial s}(0, 0) - \frac{\partial \psi}{\partial s}(T, 0) \right) = \|\nabla \log p_T(\boldsymbol{x}_T)\|^2 + \int_T^0 \frac{\partial^2 \psi}{\partial t \partial s}(t, 0) dt, \tag{64}$$

where we applied the fundamental theorem of calculus. We arrived at a seemingly more complex formula. However, we can now swap the order of the derivatives (assuming that $\log p \in C^2(\mathbb{R} \times \mathbb{R}^D)$ and $\boldsymbol{u} \in C^1(\mathbb{R} \times \mathbb{R}^D)$):

$$\frac{\partial^2 \psi}{\partial t \partial s}(t, s) = \frac{\partial^2 \psi}{\partial s \partial t}(t, s) = \frac{\partial}{\partial s} \left( \frac{\partial \psi}{\partial t}(t, s) \right). \tag{65}$$

$\frac{\partial \psi}{\partial t}$ is given by Eq. 3:

$$\frac{\partial \psi}{\partial t}(t, s) = \frac{d}{dt} \log p_t(\boldsymbol{x}_t(c(s))) = -\operatorname{div} \boldsymbol{u}_t(\boldsymbol{x}_t(c(s))) \tag{66}$$

and by the chain rule and denoting $\nabla \operatorname{div} \boldsymbol{u}_t$ the gradient of the scalar function $\boldsymbol{x} \mapsto \operatorname{div} \boldsymbol{u}_t(\boldsymbol{x})$:

$$\frac{\partial^2 \psi}{\partial s \partial t}(t, s) = \frac{d}{ds} - \operatorname{div} \boldsymbol{u}_t(\boldsymbol{x}_t(c(s))) \tag{67}$$

$$= -\left( \nabla \operatorname{div} \boldsymbol{u}_t(\boldsymbol{x}_t(c(s))) \right)^T \frac{d}{ds}(\boldsymbol{x}_t(c(s))) \tag{68}$$

$$= -\left( \nabla \operatorname{div} \boldsymbol{u}_t(\boldsymbol{x}_t(c(s))) \right)^T \frac{\partial \boldsymbol{x}_t}{\partial \boldsymbol{x}_0}(c(s)) c'(s) \tag{69}$$

$$= -\left( \nabla \operatorname{div} \boldsymbol{u}_t(\boldsymbol{x}_t(c(s))) \right)^T \frac{\partial \boldsymbol{x}_t}{\partial \boldsymbol{x}_0}(c(s)) \nabla \log p_T(c(s)). \tag{70}$$

After setting $s = 0$ we get

$$\frac{\partial^2 \psi}{\partial s \partial t}(t, 0) = -\left( \nabla \operatorname{div} \boldsymbol{u}_t(\boldsymbol{x}_t(\boldsymbol{x}_T)) \right)^T \underbrace{\frac{\partial \boldsymbol{x}_t}{\partial \boldsymbol{x}_0}(\boldsymbol{x}_T) \nabla \log p_T(\boldsymbol{x}_T)}_{=\boldsymbol{v}_t(\boldsymbol{x}_T)} \tag{71}$$

$$= -\left( \nabla \operatorname{div} \boldsymbol{u}_t(\boldsymbol{x}_t(\boldsymbol{x}_T)) \right)^T \boldsymbol{v}_t(\boldsymbol{x}_T) \tag{72}$$

$$\overset{(34)}{=} -\operatorname{div}\left( \frac{\partial \boldsymbol{u}_t}{\partial \boldsymbol{x}}(\boldsymbol{x}_t) \boldsymbol{v}_t \right), \tag{73}$$

where $\boldsymbol{x}_t = \boldsymbol{x}_t(\boldsymbol{x}_T)$ and $\boldsymbol{v}_t = \boldsymbol{v}_t(\boldsymbol{x}_T)$. After plugging into Eq. 64:

$$\frac{\partial \psi}{\partial s}(0, 0) = \|\nabla \log p_T(\boldsymbol{x}_T)\|^2 + \int_T^0 -\operatorname{div}\left( \frac{\partial \boldsymbol{u}_t}{\partial \boldsymbol{x}}(\boldsymbol{x}_t) \boldsymbol{v}_t \right) dt \tag{74}$$

which after plugging into Eq. 62 becomes

$$\nabla \log p_0(\boldsymbol{x}_0)^T \frac{\partial \boldsymbol{x}_0}{\partial \boldsymbol{x}_T} \nabla \log p_T(\boldsymbol{x}_T) = \|\nabla \log p_T(\boldsymbol{x}_T)\|^2 + \int_T^0 -\operatorname{div}\left( \frac{\partial \boldsymbol{u}_t}{\partial \boldsymbol{x}}(\boldsymbol{x}_t) \boldsymbol{v}_t \right) dt \tag{75}$$

and crucially, the score function for $t < T$ does not appear in the RHS, which can be estimated purely from derivatives of $\boldsymbol{u}_t$.

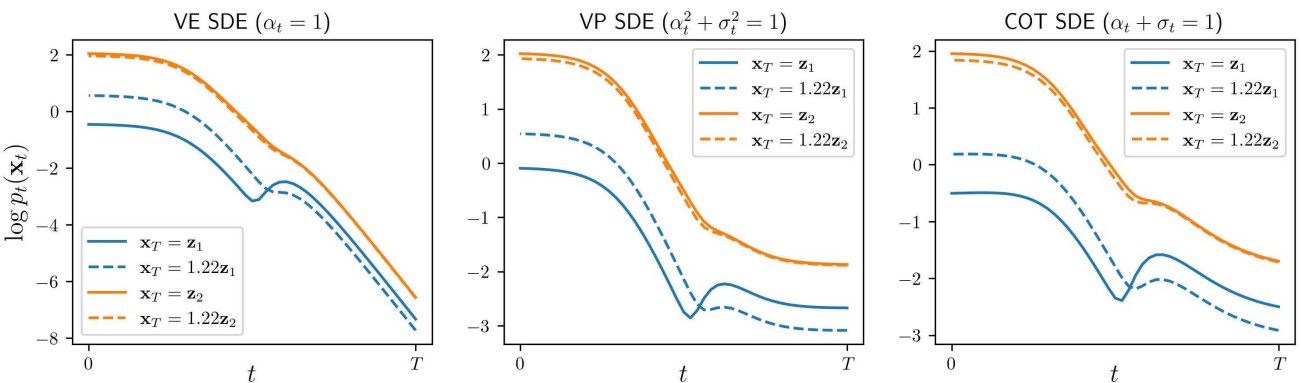

*Figure 11.* **Prior Guidance is ineffective due to unsatisfied SA condition.** For some latent codes: $\boldsymbol{x}_T = \sigma_T \mathbf{z}_1$ scaling the latent code by 1.22 increases $\log p_0(\boldsymbol{x}_0)$, while for others the same scaling leads to a decrease in $\log p_0(\boldsymbol{x}_0)$ ($\boldsymbol{x}_T = \sigma_T \mathbf{z}_1$). It can be seen that the blue lines cross, while the orange lines do not. The same behavior was observed regardless of which SDE was used.

### C.2. Score alignment holds in linear models

In this subsection, we show that the score alignment Eq. 11 holds whenever $\boldsymbol{u}_t(\boldsymbol{x}_t)$ is linear in $\boldsymbol{x}$. Such models are for example linear-drift diffusion models with Gaussian data distribution $p_0$. Score alignment is then an immediate consequence of Eq. 75. Specifically, when $\boldsymbol{u}_t$ is linear in $\boldsymbol{x}$, then $\frac{\partial \boldsymbol{u}_t}{\partial \boldsymbol{x}} \boldsymbol{v}_t$ does not depend on $\boldsymbol{x}$ and thus

$$\operatorname{div}\left(\frac{\partial \boldsymbol{u}_t}{\partial \boldsymbol{x}} \boldsymbol{v}_t\right) = 0 \tag{76}$$

and

$$\nabla \log p_0(\boldsymbol{x}_0)^T \frac{\partial \boldsymbol{x}_0}{\partial \boldsymbol{x}_T} \nabla \log p_T(\boldsymbol{x}_T) = \|\nabla \log p_T(\boldsymbol{x}_T)\|^2 + \int_T^0 0 dt = \|\nabla \log p_T(\boldsymbol{x}_T)\|^2 \geq 0. \tag{77}$$

### C.3. SA does not always hold

We provide a simple example, for which the Score Alignment condition fails and thus Prior Guidance does not lead to monotonic changes in $\log p_0(\boldsymbol{x}_0)$. We study a 2-dimensional Gaussian mixture distribution with three components: $p_0 = \frac{1}{3} \sum_{i=1}^3 \mathcal{N}(\mu_i, 0.005 \boldsymbol{I}_2)$, where $\mu_1 = [-0.3502, -0.6207]^T$, $\mu_2 = [-0.4828, 1.0680]^T$ and $\mu_3 = [-0.7789, 0.7565]^T$ ($\mu_i$ we randomly chosen).

We found two latent codes $\mathbf{z}_1 = [1.3166, -0.2252]^T$ and $\mathbf{z}_2 = [-0.1504, -0.2165]^T$ exhibiting inconsistent behaviour. Specifically, when $\boldsymbol{x}_T = \sigma_T \mathbf{z}_1$, scaling up by 1.22 *decreases* $\log p_0(\boldsymbol{x}_0)$, while for $\boldsymbol{x}_T = \sigma_T \mathbf{z}_2$ the same scaling *increases* $\log p_0(\boldsymbol{x}_0)$. We visualize this in Fig. 11 with solid lines corresponding to decoding $\boldsymbol{x}_T$ and the dashed lines the decodings of $1.22 \boldsymbol{x}_T$. This behavior was consistent regardless of which SDE was used.

## D. Derivation of Density Guidance

In this section we derive Eq. 16. From Eq. 15, we see that if $\boldsymbol{x}_t$ is following a trajectory given by $d\boldsymbol{x}_t = \tilde{\boldsymbol{u}}_t(\boldsymbol{x}_t) dt$, then

$$\frac{d \log p_t(\boldsymbol{x}_t)}{dt} = b_t(\boldsymbol{x}_t) \quad \Leftrightarrow \quad \nabla \log p_t(\boldsymbol{x}_t)^T (\tilde{\boldsymbol{u}}_t(\boldsymbol{x}_t) - \boldsymbol{u}_t(\boldsymbol{x}_t)) = b_t(\boldsymbol{x}_t) + \operatorname{div} \boldsymbol{u}_t(\boldsymbol{x}_t) \tag{78}$$

$$\Leftrightarrow \quad \tilde{\boldsymbol{u}}_t(\boldsymbol{x}_t)^T \nabla \log p_t(\boldsymbol{x}_t) = b_t(\boldsymbol{x}_t) + \operatorname{div} \boldsymbol{u}_t(\boldsymbol{x}_t) + \boldsymbol{u}_t(\boldsymbol{x}_t)^T \nabla \log p_t(\boldsymbol{x}_t) \tag{79}$$

Whenever $\nabla \log p_t(\boldsymbol{x}_t) = \boldsymbol{0}$, then RHS is satisfied only when $b_t(\boldsymbol{x}_t) = -\operatorname{div} \boldsymbol{u}_t(\boldsymbol{x}_t)$. In other words, when the score function vanishes, the infinitesimal change in $\log p_t(\boldsymbol{x}_t)$ is the same and equal to $-\operatorname{div} \boldsymbol{u}_t(\boldsymbol{x}_t)$ regardless of the choice of $\tilde{\boldsymbol{u}}_t$.

Assume now that $\nabla \log p_t(\boldsymbol{x}_t) \neq \boldsymbol{0}$. For fixed $(t, \boldsymbol{x}_t)$, we can treat the condition in Eq. 78 as a linear equation with $\boldsymbol{w} := \tilde{\boldsymbol{u}}_t(\boldsymbol{x}_t)$ being the unknown quantity we want to solve for. It is a single equation with $D$ variables (dimensionality of $\boldsymbol{w}$), i.e., it does not have a unique solution. We can choose one that satisfies additional criteria out of all possible

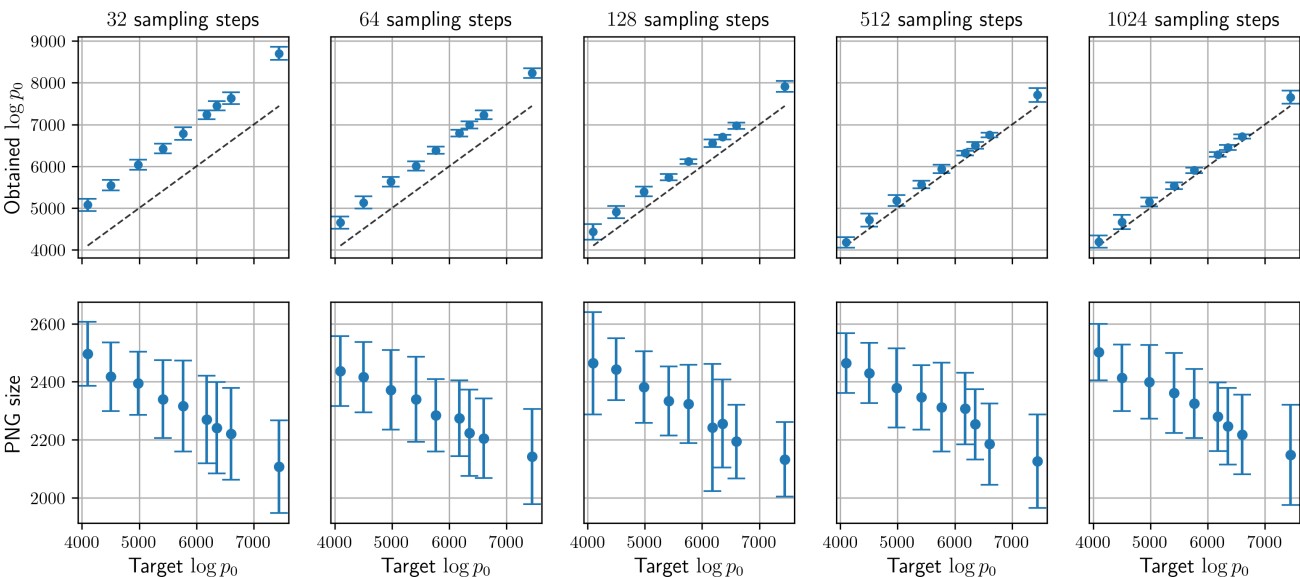

*Figure 12.* **Explicit Quantile Matching achieves exact likelihoods when step size goes to zero.** For all numbers of sampling steps, the correlation between the desired $\log p_0$ and the obtained $\log p_0$ is above 99%.

solutions. Specifically, we choose a solution that diverges from the original trajectory $\boldsymbol{u}_t(\boldsymbol{x}_t)$ the least. We therefore solve the following constrained optimization problem

$$
\min_{\boldsymbol{w} \in \mathbb{R}^D} \frac{1}{2} \|\boldsymbol{w} - \boldsymbol{u}_t(\boldsymbol{x}_t)\|^2
$$
$$
\text{s.t.} \quad \boldsymbol{w}^T \nabla \log p_t(\boldsymbol{x}_t) = b_t(\boldsymbol{x}_t) + \operatorname{div} \boldsymbol{u}_t(\boldsymbol{x}_t) + \boldsymbol{u}_t(\boldsymbol{x}_t)^T \nabla \log p_t(\boldsymbol{x}_t), \tag{80}
$$

which is treated in Appendix A.1. The solution is

$$
\boldsymbol{w} = \boldsymbol{u}_t(\boldsymbol{x}_t) + \frac{b_t(\boldsymbol{x}_t) + \operatorname{div} \boldsymbol{u}_t(\boldsymbol{x}_t) + \boldsymbol{u}_t(\boldsymbol{x}_t)^T \nabla \log p_t(\boldsymbol{x}_t) - \nabla \log p_t(\boldsymbol{x}_t)^T \boldsymbol{u}_t(\boldsymbol{x}_t)}{\|\nabla \log p_t(\boldsymbol{x}_t)\|^2} \nabla \log p_t(\boldsymbol{x}_t) \tag{81}
$$

$$
= \boldsymbol{u}_t(\boldsymbol{x}_t) + \frac{b_t(\boldsymbol{x}_t) + \operatorname{div} \boldsymbol{u}_t(\boldsymbol{x}_t)}{\|\nabla \log p_t(\boldsymbol{x}_t)\|^2} \nabla \log p_t(\boldsymbol{x}_t), \tag{82}
$$

which matches Eq. 16.

## E. Explicit quantile matching

To demonstrate claims made in Section 4.1, we performed density guidance with explicit quantile matching on CIFAR-10. Specifically, we estimated the quantile function $\phi_t$ as described in Section 4.1 by sampling $K^3$ samples $\boldsymbol{x}_T \sim p_T$ and solving the PF-ODE (Eq. 6) from $t = T$ to $t = 0$ in 1024 Euler steps. For all samples, we estimated the marginal log-density at each step $\log p_t(\boldsymbol{x}_t)$ with Eq. 3 and defined the quantile function $\phi_t$ as empirical quantiles of $\log p_t(\boldsymbol{x}_t)$. We then define $b_t(\boldsymbol{x}) = \frac{d}{dt}\phi_t$, which we estimate with a moving average of finite difference estimates.

We found that the difference between the desired values of log-density and the obtained ones goes to zero as we decrease the discretization error (increase the number of sampling steps). Interestingly, for lower number of sampling steps, even though we do not obtain exact desired values of likelihood, the correlation between the desired values and the obtained ones remains above 99%, even for as few as 32 Euler sampling steps. This means that for all values of the number of sampling steps, we saw a monotonic relationship between the target $\log p_0$ and the amount of detail (PNG size). Please see Fig. 12. As "ground truth" $\log p_0(\boldsymbol{x}_0)$ estimate, we used Eq. 3 for encoding $\boldsymbol{x}_0$ to $\boldsymbol{x}_T$ with the PF-ODE (Eq. 6) in 1024 Euler steps.

---

[3]We tested $K = [16, 32, 64, 128, 256, 512, 1024]$ and found that using $K = 128$ is enough to ensure a correlation between the desired value of log-density and the obtained one is above 99%.

# F. Asymptotic behaviour of $\Delta \log p(\boldsymbol{x}) + \|\nabla \log p_t(\boldsymbol{x})\|^2$

In this section, we discuss an observation that proved useful in determining *typical* values $\frac{d \log p_t(\boldsymbol{x}_t)}{dt}$ in diffusion models. Specifically, we observed that for some distributions $p_0$ after diffusing into $p_t$ via the forward process $p(\boldsymbol{x}_t|\boldsymbol{x}_0) = \mathcal{N}(\alpha_t \boldsymbol{x}_0, \sigma_t^2 \boldsymbol{I}_D)$, the following holds

$$h(\boldsymbol{x}) = \frac{\sigma_t^2 \left( \Delta \log p_t(\boldsymbol{x}) + \|\nabla \log p_t(\boldsymbol{x})\|^2 \right)}{\sqrt{2D}} \xrightarrow[D\to\infty]{d} \mathcal{N}(0,1), \tag{83}$$

where $D$ denotes the dimension of the distribution $p_t$ and "$\xrightarrow{d}$" denotes convergence in distribution.

## F.1. Single data point

We begin by showing Eq. 83 for the simplest possible case, where $p_0 = \delta_{\boldsymbol{x}_0}$. In that case $p_t = \mathcal{N}(\alpha_t \boldsymbol{x}_0, \sigma_t^2 \boldsymbol{I}_D)$ and

$$\nabla \log p_t(\boldsymbol{x}) = \frac{\alpha_t \boldsymbol{x}_0 - \boldsymbol{x}}{\sigma_t^2}$$

$$\Delta \log p_t(\boldsymbol{x}) = -\frac{D}{\sigma_t^2}.$$

Since $\boldsymbol{x} = \alpha_t \boldsymbol{x}_0 + \sigma_t \varepsilon$ for $\varepsilon \sim \mathcal{N}(\boldsymbol{0}, \boldsymbol{I}_D)$ and our expression becomes

$$h(\boldsymbol{x}) = \frac{\sigma_t^2 \left( -\frac{D}{\sigma_t^2} + \frac{1}{\sigma_t^2} \|\varepsilon\|^2 \right)}{\sqrt{2D}} = \frac{\sum_j (\varepsilon_j^2 - 1)}{\sqrt{2D}}. \tag{84}$$

Since $\{\varepsilon_j^2\}_j$ are i.i.d. random variables with $\chi_1^2$ distribution, we have that $\mathbb{E}[\varepsilon_j^2] = 1$, $\mathrm{Var}[\varepsilon_j^2] = 2$, and the claim follows from the central limit theorem.

## F.2. Non-isotropic Gaussian distribution

When $p_t$ is Gaussian, but with non-diagonal covariance an analogous result holds when the covariance matrix satisfies some additional conditions. We begin with a useful lemma.

**Lemma F.1** (Quadratic CLT (de Jong, 1987)). *Suppose* $A = [a_{ij}] \in \mathbb{R}^{D \times D}$ *is a real symmetric matrix with eigenvalues* $\lambda_1, \ldots, \lambda_D$. *Let* $\{\varepsilon_j\}_{j=1\ldots D}$ *be independent variables such that* $\varepsilon_j \sim \mathcal{N}(0,1)$.

*If*

$$\lim_{D\to\infty} \frac{\max_{j\le D} \lambda_j^2}{\sum_{j\le D} \lambda_j^2} = 0, \tag{85}$$

*then*

$$\frac{\varepsilon^T A \varepsilon - \mathrm{Tr}(A)}{\sqrt{2}\|A\|_F} \xrightarrow[D\to\infty]{d} \mathcal{N}(0,1). \tag{86}$$

Let $p_t = \mathcal{N}(\mu, \Sigma)$ for $\Sigma$ satisfying the following conditions. Denoting $\Sigma = LL^T$, and $\tilde{\Sigma} = L^{-1}(L^T)^{-1}$ with $\lambda_1, \ldots, \lambda_D$ eigenvalues of $\tilde{\Sigma}$, we assume

$$\lim_{D\to\infty} \frac{\max_{j\le D} \lambda_j^2}{\sum_{j\le D} \lambda_j^2} = 0 \tag{87}$$

Note that for $\Sigma = \sigma_t^2 \boldsymbol{I}_D$, we have $\tilde{\Sigma} = \frac{1}{\sigma_t^2} \boldsymbol{I}_D$, $\lambda_k = \frac{1}{\sigma_t^2}$, and all the above conditions becomes $\lim_{D\to\infty} \frac{1}{D} = 0$, which of course holds. Then

$$h(\boldsymbol{x}) = \frac{\Delta \log p_t(\boldsymbol{x}) + \|\nabla \log p_t(\boldsymbol{x})\|^2}{\sqrt{2}\|\nabla^2 \log p_t(\boldsymbol{x})\|_F} \xrightarrow[D\to\infty]{d} \mathcal{N}(0,1). \tag{88}$$

For $\boldsymbol{x} \sim \mathcal{N}(\mu, \Sigma)$, we can represent $\boldsymbol{x} = \mu + L\varepsilon$ for $\Sigma = LL^T$ and $\varepsilon \sim \mathcal{N}(\boldsymbol{0}, \boldsymbol{I}_D)$. In this case, we have

$$\nabla \log p_t(\boldsymbol{x}) = \Sigma^{-1}(\mu - \boldsymbol{x}) = -(L^T)^{-1}\varepsilon$$

$$\nabla^2 \log p_t(\boldsymbol{x}) = -\Sigma^{-1}$$

$$\Delta \log p_t(\boldsymbol{x}) = -\mathrm{Tr}(\Sigma^{-1}).$$

Note that $\|\nabla \log p_t(\boldsymbol{x})\|^2 = \boldsymbol{\varepsilon}^T \tilde{\Sigma} \boldsymbol{\varepsilon}$, where $\tilde{\Sigma} = L^{-1}(L^T)^{-1}$. Since $\Sigma^{-1} = (L^T)^{-1} L^{-1}$ and $\text{Tr}(AB) = \text{Tr}(BA)$, we have $\Delta \log p_t(\boldsymbol{x}) = -\text{Tr}(\tilde{\Sigma})$ and $\|\tilde{\Sigma}\|_F = \|\Sigma^{-1}\|_F = \|\nabla^2 \log p_t(\boldsymbol{x})\|_F$ We can now write

$$h(\boldsymbol{x}) = \frac{\Delta \log p_t(\boldsymbol{x}) + \|\nabla \log p_t(\boldsymbol{x})\|^2}{\sqrt{2}\|\nabla^2 \log p_t(\boldsymbol{x})\|_F} = \frac{\boldsymbol{\varepsilon}^T \tilde{\Sigma} \boldsymbol{\varepsilon} - \text{Tr}(\tilde{\Sigma})}{\sqrt{2}\|\tilde{\Sigma}\|_F} \xrightarrow[D \to \infty]{d} \mathcal{N}(0, 1) \tag{89}$$

from Lemma F.1.

### F.3. Gaussian Mixture

Usually, the distributions we are interested in can be represented as $p_0 = \frac{1}{K} \sum_{k=1}^{K} \delta_{\boldsymbol{x}_k}$, where $\{\boldsymbol{x}_k\}_k \subset \mathbb{R}^D$ is the data set. We show that in this case Eq. 83 also holds. In that case, $p_t = \frac{1}{K} \sum_{k=1}^{K} \mathcal{N}(\mu_k, \sigma_t^2 \boldsymbol{I}_D)$, where $\mu_k = \alpha_t \boldsymbol{x}_k$. We will use the following identity, which holds for any $p(\boldsymbol{x})$:

$$\Delta \log p(\boldsymbol{x}) + \|\nabla \log p(\boldsymbol{x})\|^2 = \frac{\Delta p(\boldsymbol{x})}{p(\boldsymbol{x})}. \tag{90}$$

In the Gaussian mixture case (denoting $p_k = \mathcal{N}(\mu_k, \sigma_t^2 \boldsymbol{I}_D)$), we have

$$\frac{\partial}{\partial x^i} p_t(\boldsymbol{x}) = \frac{1}{K} \sum_k p_k(\boldsymbol{x}) \frac{\mu_k^i - x^i}{\sigma_t^2} = \frac{1}{K \sigma_t^2} \sum_k p_k(\boldsymbol{x})(\mu_k^i - x^i)$$

and

$$\frac{\partial^2}{\partial (x^i)^2} p_t(\boldsymbol{x}) = \frac{1}{K \sigma_t^2} \sum_k \frac{\partial}{\partial x^i} p_k(\boldsymbol{x})(\mu_k^i - x^i) - \frac{1}{\sigma_t^2} p_t(\boldsymbol{x}) = \frac{1}{K \sigma_t^4} \sum_k p_k(\boldsymbol{x})(\mu_k^i - x^i)^2 - \frac{1}{\sigma_t^2} p_t(\boldsymbol{x}).$$

Therefore, we have

$$h(\boldsymbol{x}) = \frac{\sigma_t^2 \left( \Delta \log p_t(\boldsymbol{x}) + \|\nabla \log p_t(\boldsymbol{x})\|^2 \right)}{\sqrt{2D}} = \frac{\sigma_t^2 \Delta p_t(\boldsymbol{x})}{p_t(\boldsymbol{x})\sqrt{2D}} = \frac{\sum_k w_k(\boldsymbol{x}) \|\frac{\boldsymbol{x} - \mu_k}{\sigma_t}\|^2 - D}{\sqrt{2D}}, \tag{91}$$

where $w_k(\boldsymbol{x}) := \frac{p_k(\boldsymbol{x})}{p(\boldsymbol{x})}$. In Theorem 1 we show that $h(\boldsymbol{x}) \xrightarrow{d} N(0, 1)$. We additionally verify this hypothesis numerically. Specifically, we set the number of components to $K = 128$ and sample $\{\mu_k\}$ from $\mathcal{N}(\boldsymbol{0}, \sigma_t^2 \boldsymbol{I}_D)$. We then sample $N = 16384$ samples $\boldsymbol{x}_j \sim p_t(\boldsymbol{x})$ and evaluate corresponding values of $h(\boldsymbol{x})$ with Eq. 91. We repeat this experiment for three values of $\sigma_t \in \{0.5, 1, 10\}$. To test whether the distribution of $h(\boldsymbol{x})$ approaches $\mathcal{N}(0, 1)$ for larger $D$, we repeat this experiment for $D = 2^m$ for $m = 6, 7, \ldots, 12$ and evaluate the p-value of a normality test on $h(\boldsymbol{x})$ [4]. We see that for $D$ greater than $\approx 1000$, the distribution of $h(\boldsymbol{x})$ is close to $\mathcal{N}(0, 1)$ as evidenced by p-value being greater than the commonly used significance threshold $\alpha = 0.05$. Please see Fig. 13.

### F.4. Image data

In this section we study $p_0$ being CIFAR-10 image data distribution and $\nabla \log p_t(\boldsymbol{x})$ being approximated with a neural network. Specifically, we uniformly sample different times $t \in (0, T]$ and corresponding noisy samples $\boldsymbol{x}_t \sim p_t(\boldsymbol{x})$. Then, we estimate $\nabla \log p_t(\boldsymbol{x})$ using a model and $\Delta \log p_t(\boldsymbol{x}) \approx \text{div} \nabla \log p_t(\boldsymbol{x})$ using the Hutchinson's trick. Finally, we plot $\Phi^{-1}(h(\boldsymbol{x}_t))$ for $h(\boldsymbol{x}_t)$ estimated using Eq. 83, where $\Phi$ is the cumulative density function of $\mathcal{N}(0, 1)$. If $h(\boldsymbol{x}_t) \sim \mathcal{N}(0, 1)$, then $\Phi^{-1}(h(\boldsymbol{x}_t)) \sim \mathcal{U}(0, 1)$ for all $t \in (0, T]$. Indeed, this is precisely the observed behaviour for two different choices of the forward process: $\alpha_t^2 + \sigma_t^2 = 1$ (VP-SDE) and $\alpha_t + \sigma_t = 1$ (CFM) confirming that this finding also holds for high dimensional image data. See Fig. 14.

## G. Stochastic density guidance

In this section, we derive the stochastic density guidance method. A central tool in this section is Itô's lemma (Itô, 1951), a generalization of the total derivative to stochastic processes.

---

[4] https://docs.scipy.org/doc/scipy/reference/generated/scipy.stats.normaltest.html

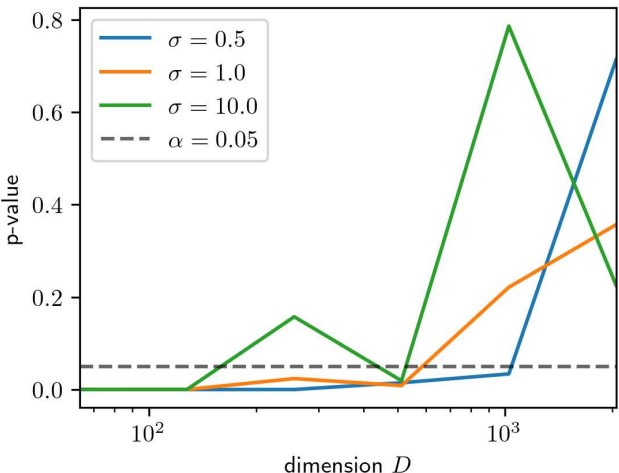

*Figure 13.* $h(\boldsymbol{x})$ **approaches** $\mathcal{N}(0,1)$ **for larger** $D$.

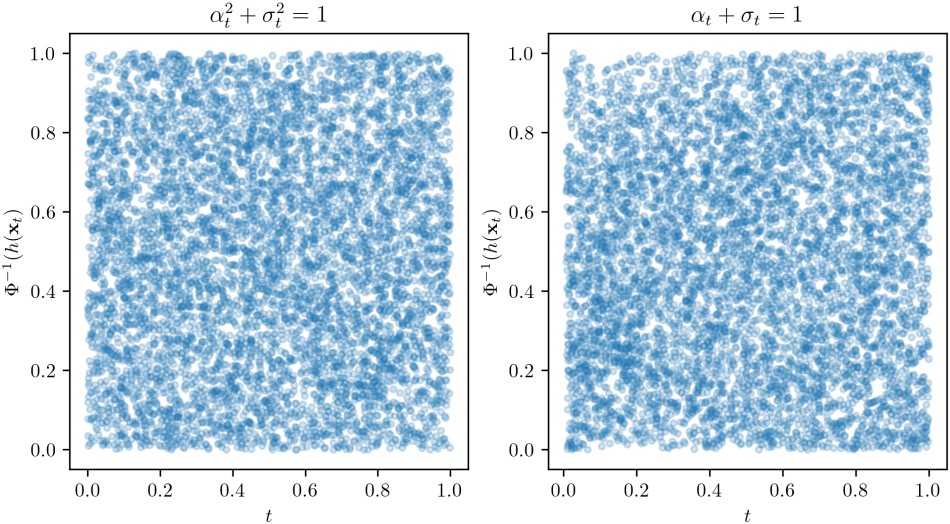

*Figure 14.* $h(\boldsymbol{x}_t)$ **approximately follows** $\mathcal{N}(0,1)$ **for various definitions of the diffusion process.**

**Lemma G.1** (Itô's Lemma). *Let $d\boldsymbol{x}_t = \mu(t, \boldsymbol{x}_t)dt + \boldsymbol{G}(t, \boldsymbol{x}_t)d\mathrm{W}_t$ be a D-dimensional Itô process with $\mu : \mathbb{R} \times \mathbb{R}^D \to \mathbb{R}^D$, $\boldsymbol{G} : \mathbb{R} \times \mathbb{R}^D \to \mathbb{R}^{D \times D}$ and $\mathrm{W}$ the Wiener process in $\mathbb{R}^D$. For a smooth function $h : \mathbb{R} \times \mathbb{R}^D \to \mathbb{R}$, it holds that $h(t, \boldsymbol{x}_t)$ is also an Itô process with the following dynamics*

$$dh(t, \boldsymbol{x}_t) = \left( \frac{\partial h}{\partial t}(t, \boldsymbol{x}_t) + \mu(t, \boldsymbol{x}_t)^T \frac{\partial h}{\partial \boldsymbol{x}}(t, \boldsymbol{x}_t) + \frac{1}{2}\mathrm{Tr}\left( \boldsymbol{G}(t, \boldsymbol{x}_t)^T \nabla^2 h(t, \boldsymbol{x}_t)\boldsymbol{G}(t, \boldsymbol{x}_t) \right) \right) dt$$
$$+ \frac{\partial h}{\partial \boldsymbol{x}}(t, \boldsymbol{x}_t)^T \boldsymbol{G}(t, \boldsymbol{x}_t)d\mathrm{W}_t, \tag{92}$$

*where $\nabla^2 h$ is the Hessian matrix of $h$ w.r.t $\boldsymbol{x}$. In the case when $\boldsymbol{G}(t, \boldsymbol{x}) = \varphi(t)\boldsymbol{I}_D$, the dynamics simplify to*

$$dh(t, \boldsymbol{x}_t) = \left( \frac{\partial h}{\partial t}(t, \boldsymbol{x}_t) + \mu(t, \boldsymbol{x}_t)^T \frac{\partial h}{\partial \boldsymbol{x}}(t, \boldsymbol{x}_t) + \frac{1}{2}\varphi^2(t)\Delta_{\boldsymbol{x}} h(t, \boldsymbol{x}_t) \right) ds + \varphi(t) \frac{\partial h}{\partial \boldsymbol{x}}(t, \boldsymbol{x}_t)^T d\mathrm{W}_t \tag{93}$$

In contrast to the total derivative for deterministic processes, the forward and reverse-time dynamics are not the same for stochastic processes. We now prove the reverse-time Itô's lemma, which will be useful in our derivation as our convention is that sampling happens backward in time, from $t = T$ to $t = 0$.

**Corollary 1** (Reverse-time Itô's lemma). *Let $d\boldsymbol{x}_t = \mu(t, \boldsymbol{x}_t)dt + \boldsymbol{G}(t, \boldsymbol{x}_t)d\overline{\mathrm{W}}_t$, $dt < 0$, $\overline{\mathrm{W}}$ the Wiener process running backwards in time from $t = T$ to $t = 0$ and $\mu$ and $G$ are as in Lemma G.1. Then*

$$dh(t, \boldsymbol{x}_t) = \left( \frac{\partial h}{\partial t}(t, \boldsymbol{x}_t) + \mu(t, \boldsymbol{x}_t)^T \frac{\partial h}{\partial \boldsymbol{x}}(t, \boldsymbol{x}_t) - \frac{1}{2}\mathrm{Tr}\left( \boldsymbol{G}(t, \boldsymbol{x}_t)^T \nabla^2 h(t, \boldsymbol{x}_t)\boldsymbol{G}(t, \boldsymbol{x}_t) \right) \right) dt$$
$$+ \frac{\partial h}{\partial \boldsymbol{x}}(t, \boldsymbol{x}_t)^T \boldsymbol{G}(t, \boldsymbol{x}_t)d\overline{\mathrm{W}}_t, \tag{94}$$

*with the modifications coming from time-reversal highlighted in blue.*

*Proof.* Let $s = T - t$. Since $ds = -dt$, the dynamics of $\boldsymbol{x}$ can be equivalently written as (Dockhorn et al., 2022):

$$d\boldsymbol{x}_s = -\mu(T - s, \boldsymbol{x}_s)ds + \boldsymbol{G}(T - s, \boldsymbol{x}_s)d\mathrm{W}_s \tag{95}$$

for the standard Wiener process W and $ds > 0$. Now let $\tilde{h}(s, \boldsymbol{x}) := h(T - s, \boldsymbol{x})$. Applying Itô's lemma to $\tilde{h}$ yields

$$d\tilde{h}(s, \boldsymbol{x}_s) = \left( \frac{\partial \tilde{h}}{\partial s}(s, \boldsymbol{x}_s) - \mu(T - s, \boldsymbol{x}_s)^T \frac{\partial \tilde{h}}{\partial \boldsymbol{x}}(s, \boldsymbol{x}_s) + \frac{1}{2}\mathrm{Tr}\left( \boldsymbol{G}(T - s, \boldsymbol{x}_s)^T \nabla^2 \tilde{h}(s, \boldsymbol{x}_s)\boldsymbol{G}(T - s, \boldsymbol{x}_s) \right) \right) ds$$
$$+ \frac{\partial \tilde{h}}{\partial \boldsymbol{x}}(s, \boldsymbol{x}_s)^T \boldsymbol{G}(T - s, \boldsymbol{x}_s)d\mathrm{W}_s$$
$$= \left( -\frac{\partial h}{\partial t}(T - s, \boldsymbol{x}_s) - \mu(T - s, \boldsymbol{x}_s)^T \frac{\partial h}{\partial \boldsymbol{x}}(T - s, \boldsymbol{x}_s) + \frac{1}{2}\mathrm{Tr}\left( \boldsymbol{G}(T - s, \boldsymbol{x}_s)^T \nabla^2 h(T - s, \boldsymbol{x}_s)\boldsymbol{G}(T - s, \boldsymbol{x}_s) \right) \right) ds$$
$$+ \frac{\partial h}{\partial \boldsymbol{x}}(T - s, \boldsymbol{x}_s)^T \boldsymbol{G}(T - s, \boldsymbol{x}_s)d\mathrm{W}_s. \tag{96}$$

The claim follows from switching back to running backward in time $t \leftarrow T - s$ $\qquad \square$

Recall Eq. 7 which describes stochastic sampling from a CNF model

$$d\boldsymbol{x}_t = \left( \boldsymbol{u}_t(\boldsymbol{x}_t) - \frac{1}{2}\varphi^2(t)\nabla \log p_t(\boldsymbol{x}_t) \right) dt + \varphi(t)d\overline{\mathrm{W}}_t \tag{7}$$

for which, we know how the log-density evolves (Eq. 8):

$$d\log p_t(\boldsymbol{x}_t) = \left( -\mathrm{div}\, \boldsymbol{u}_t(\boldsymbol{x}) - \frac{1}{2}\varphi^2(t)\left( \Delta \log p_t(\boldsymbol{x}) + \frac{1}{2}\varphi^2(t)\|\nabla \log p_t(\boldsymbol{x})\|^2 \right) \right) dt + \varphi(t)\nabla \log p_t(\boldsymbol{x}_t)^T d\overline{\mathrm{W}}_t. \tag{8}$$

We now ask: how can modify the stochastic dynamics (Eq. 7) so that

$$d\log p_t(\boldsymbol{x}_t) = b_t(\boldsymbol{x}_t) \tag{97}$$

for some given $b_t$. Suppose that $\boldsymbol{x}$ is following

$$d\boldsymbol{x}_t = \tilde{\boldsymbol{u}}_t(\boldsymbol{x}_t)dt + \boldsymbol{G}(t, \boldsymbol{x}_t)d\overline{\mathrm{W}}_t \tag{98}$$

for some $\tilde{\boldsymbol{u}}$ and $\boldsymbol{G}$. To evaluate the change log-density we will use Corollary 1 applied to $h(t, \boldsymbol{x}) = \log p_t(\boldsymbol{x})$:

$$d\log p_t(\boldsymbol{x}_t) = \Big(\frac{\partial \log p_t}{\partial t}(\boldsymbol{x}_t) + \tilde{\boldsymbol{u}}_t(\boldsymbol{x}_t)^T \nabla \log p_t(\boldsymbol{x}_t) - \frac{1}{2}\mathrm{Tr}\left(\boldsymbol{G}(t, \boldsymbol{x}_t)^T \nabla^2 \log p_t(\boldsymbol{x}_t)\boldsymbol{G}(t, \boldsymbol{x}_t)\right)\Big)dt \tag{99}$$
$$+ \nabla \log p_t(\boldsymbol{x}_t)^T \boldsymbol{G}(t, \boldsymbol{x}_t)d\overline{\mathrm{W}}_t.$$

Since we assumed that $d\log p_t(\boldsymbol{x}_t) = b_t(\boldsymbol{x}_t)dt$, the stochastic component of $d\log p_t(\boldsymbol{x}_t)$ must vanish, i.e. $\nabla \log p_t(\boldsymbol{x})^T \boldsymbol{G}(t, \boldsymbol{x}) = \boldsymbol{0}$. There are many $\boldsymbol{G}$ that satisfy this condition including a trivial $\boldsymbol{G} \equiv \boldsymbol{0}$. However, standard stochastic sampling (Eq. 7) assumes isotropic noise, i.e. $\boldsymbol{G}(t, \boldsymbol{x}) = \varphi(t)\boldsymbol{I}_D$ and we want to match that as closely as possible. An optimal solution (Lemma A.2) to this problem is the projection $\boldsymbol{G}(t, \boldsymbol{x}) = \varphi(t)\boldsymbol{P}_t(\boldsymbol{x})$ for:

$$\boldsymbol{P}_t(\boldsymbol{x}) = \boldsymbol{I}_D - \left(\frac{\nabla \log p_t(\boldsymbol{x})}{\|\nabla \log p_t(\boldsymbol{x})\|}\right)\left(\frac{\nabla \log p_t(\boldsymbol{x})}{\|\nabla \log p_t(\boldsymbol{x})\|}\right)^T. \tag{100}$$

Clearly $\boldsymbol{P}_t(\boldsymbol{x})^T = \boldsymbol{P}_t(\boldsymbol{x})$. Furthermore, since $\boldsymbol{P}_t$ is a projection matrix, it also holds that $\boldsymbol{P}_t(\boldsymbol{x})\boldsymbol{P}_t(\boldsymbol{x}) = \boldsymbol{P}_t(\boldsymbol{x})$. Now we can plug this into Eq. 99 and we obtain

$$d\log p_t(\boldsymbol{x}_t) = \Big(\frac{\partial \log p_t}{\partial t}(\boldsymbol{x}_t) + \tilde{\boldsymbol{u}}_t(\boldsymbol{x}_t)^T \nabla \log p_t(\boldsymbol{x}_t) - \frac{1}{2}\varphi^2(t)\mathrm{Tr}\left(\boldsymbol{P}_t(\boldsymbol{x}_t)^T \nabla^2 \log p_t(\boldsymbol{x}_t)\boldsymbol{P}_t(\boldsymbol{x}_t)\right)\Big)dt. \tag{101}$$

Using the symmetry and idempotency of $P_t$, and properties of the trace (linearity and $\mathrm{Tr}(AB) = \mathrm{Tr}(BA)$), we have

$$\mathrm{Tr}\left(\boldsymbol{P}_t(\boldsymbol{x})^T \nabla^2 \log p_t(\boldsymbol{x})\boldsymbol{P}_t(\boldsymbol{x})\right) = \mathrm{Tr}\left(\boldsymbol{P}_t(\boldsymbol{x})\nabla^2 \log p_t(\boldsymbol{x})\right) \tag{102}$$

$$= \mathrm{Tr}\left(\nabla^2 \log p_t(\boldsymbol{x})\right) - \frac{1}{\|\nabla \log p_t(\boldsymbol{x})\|^2}\mathrm{Tr}\left(\nabla \log p_t(\boldsymbol{x})\nabla \log p_t(\boldsymbol{x})^T \nabla^2 \log p_t(\boldsymbol{x})\right) \tag{103}$$

$$= \Delta \log p_t(\boldsymbol{x}) - \frac{\nabla \log p_t(\boldsymbol{x})^T \nabla^2 \log p_t(\boldsymbol{x})\nabla \log p_t(\boldsymbol{x})}{\|\nabla \log p_t(\boldsymbol{x})\|^2} \tag{104}$$

$$= \Delta \log p_t(\boldsymbol{x}) - \mathcal{R}(\nabla^2 \log p_t(\boldsymbol{x}), \nabla \log p_t(\boldsymbol{x})), \tag{105}$$

where

$$\mathcal{R}(\nabla^2 \log p_t(\boldsymbol{x}), \nabla \log p_t(\boldsymbol{x})) = \frac{\nabla \log p_t(\boldsymbol{x})^T \nabla^2 \log p_t(\boldsymbol{x})\nabla \log p_t(\boldsymbol{x})}{\|\nabla \log p_t(\boldsymbol{x})\|^2} \tag{106}$$

represents the Rayleigh quotient of the Hessian evaluated at $\nabla \log p_t(x)$. Furthermore, from Eq. 45, we have

$$\frac{\partial \log p_t}{\partial t}(\boldsymbol{x}) = -\mathrm{div}\,\boldsymbol{u}_t(\boldsymbol{x}) - \nabla \log p_t(\boldsymbol{x})^T \boldsymbol{u}_t(\boldsymbol{x}). \tag{107}$$

Combining these, we get

$$b_t(\boldsymbol{x}_t) = \frac{d\log p_t(\boldsymbol{x}_t)}{dt} = -\mathrm{div}\,\boldsymbol{u}_t(\boldsymbol{x}_t) + \nabla \log p_t(\boldsymbol{x}_t)^T (\tilde{\boldsymbol{u}}_t(\boldsymbol{x}_t) - \boldsymbol{u}_t(\boldsymbol{x}_t))$$
$$- \frac{1}{2}\varphi^2(t)\left(\Delta \log p_t(\boldsymbol{x}) - \mathcal{R}(\nabla^2 \log p_t(\boldsymbol{x}), \nabla \log p_t(\boldsymbol{x}))\right). \tag{108}$$

Any $\tilde{\boldsymbol{u}}_t(\boldsymbol{x})$ satisfying Eq. 108 guarantees the desired evolution of log-density. However, we wish to minimize the discrepancy from the new drift $\tilde{\boldsymbol{u}}_t(\boldsymbol{x}_t)$ and the one from Eq. 7, which guarantees sampling from the correct distribution: $\boldsymbol{u}_t(\boldsymbol{x}_t) - \frac{1}{2}\varphi^2(t)\nabla \log p_t(\boldsymbol{x}_t)$. Therefore, we solve the constrained optimization problem:

$$\min_{\tilde{\boldsymbol{u}} \in \mathbb{R}^D} \frac{1}{2}\|\tilde{\boldsymbol{u}} - \boldsymbol{u}_t(\boldsymbol{x}_t) + \frac{1}{2}\varphi^2(t)\nabla \log p_t(\boldsymbol{x}_t)\|^2 \tag{109}$$

s.t. $\tilde{\boldsymbol{u}}$ is a solution of Eq. 108.

This is a problem setting discussed and solved in Appendix A.1 with

$$\begin{cases} \boldsymbol{x} &= \tilde{\boldsymbol{u}} \\ \boldsymbol{y} &= \boldsymbol{u}_t(\boldsymbol{x}_t) - \frac{1}{2}\varphi^2(t)\nabla \log p_t(\boldsymbol{x}_t) \\ \boldsymbol{v} &= \nabla \log p_t(\boldsymbol{x}_t) \\ a &= b_t(\boldsymbol{x}_t) + \operatorname{div}\boldsymbol{u}_t(\boldsymbol{x}_t) + \boldsymbol{u}_t(\boldsymbol{x}_t)^T \nabla \log p_t(\boldsymbol{x}_t) + \frac{1}{2}\varphi^2(t)\left(\Delta \log p_t(\boldsymbol{x}) - \mathcal{R}(\nabla^2 \log p_t(\boldsymbol{x}), \nabla \log p_t(\boldsymbol{x}))\right) \end{cases} \tag{110}$$

After substituting

$$\tilde{\boldsymbol{u}}_t(\boldsymbol{x}_t) = \boldsymbol{y} + \frac{a - \boldsymbol{v}^T \boldsymbol{y}}{\|\boldsymbol{v}\|^2}\boldsymbol{v} \tag{111}$$

$$= \boldsymbol{u}_t(\boldsymbol{x}_t) - \frac{1}{2}\varphi^2(t)\nabla \log p_t(\boldsymbol{x}_t) \tag{112}$$

$$+ \frac{b_t(\boldsymbol{x}_t) + \operatorname{div}\boldsymbol{u}_t(\boldsymbol{x}_t) + \boldsymbol{u}_t(\boldsymbol{x}_t)^T \nabla \log p_t(\boldsymbol{x}_t) - \nabla \log p_t(\boldsymbol{x}_t)^T \left(\boldsymbol{u}_t(\boldsymbol{x}_t) - \frac{1}{2}\varphi^2(t)\nabla \log p_t(\boldsymbol{x}_t)\right)}{\|\nabla \log p_t(\boldsymbol{x}_t)\|^2}\nabla \log p_t(\boldsymbol{x}_t) \tag{113}$$

$$+ \frac{1}{2}\varphi^2(t)\frac{\Delta \log p_t(\boldsymbol{x}) - \mathcal{R}(\nabla^2 \log p_t(\boldsymbol{x}), \nabla \log p_t(\boldsymbol{x}))}{\|\nabla \log p_t(\boldsymbol{x}_t)\|^2}\nabla \log p_t(\boldsymbol{x}_t) \tag{114}$$

$$= \boldsymbol{u}_t(\boldsymbol{x}_t) - \frac{1}{2}\varphi^2(t)\nabla \log p_t(\boldsymbol{x}_t) + \frac{b_t(\boldsymbol{x}_t) + \operatorname{div}\boldsymbol{u}_t(\boldsymbol{x}_t) + \frac{1}{2}\varphi^2(t)\|\nabla \log p_t(\boldsymbol{x}_t)\|^2}{\|\nabla \log p_t(\boldsymbol{x}_t)\|^2}\nabla \log p_t(\boldsymbol{x}_t) \tag{115}$$

$$+ \frac{1}{2}\varphi^2(t)\frac{\Delta \log p_t(\boldsymbol{x}) - \mathcal{R}(\nabla^2 \log p_t(\boldsymbol{x}), \nabla \log p_t(\boldsymbol{x}))}{\|\nabla \log p_t(\boldsymbol{x}_t)\|^2}\nabla \log p_t(\boldsymbol{x}_t) \tag{116}$$

$$= \boldsymbol{u}_t(\boldsymbol{x}_t) - \frac{1}{2}\varphi^2(t)\nabla \log p_t(\boldsymbol{x}_t) + \frac{b_t(\boldsymbol{x}_t) + \operatorname{div}\boldsymbol{u}_t(\boldsymbol{x}_t)}{\|\nabla \log p_t(\boldsymbol{x}_t)\|^2}\nabla \log p_t(\boldsymbol{x}_t) + \frac{1}{2}\varphi^2(t)\nabla \log p_t(\boldsymbol{x}_t) \tag{117}$$

$$+ \frac{1}{2}\varphi^2(t)\frac{\Delta \log p_t(\boldsymbol{x}) - \mathcal{R}(\nabla^2 \log p_t(\boldsymbol{x}), \nabla \log p_t(\boldsymbol{x}))}{\|\nabla \log p_t(\boldsymbol{x}_t)\|^2}\nabla \log p_t(\boldsymbol{x}_t) \tag{118}$$

$$= \boldsymbol{u}_t(\boldsymbol{x}_t) + \frac{b_t(\boldsymbol{x}_t) + \operatorname{div}\boldsymbol{u}_t(\boldsymbol{x}_t) + \frac{1}{2}\varphi^2(t)\left(\Delta \log p_t(\boldsymbol{x}) - \mathcal{R}(\nabla^2 \log p_t(\boldsymbol{x}), \nabla \log p_t(\boldsymbol{x}))\right)}{\|\nabla \log p_t(\boldsymbol{x}_t)\|^2}\nabla \log p_t(\boldsymbol{x}_t). \tag{119}$$

The solution is

$$\tilde{\boldsymbol{u}}_t(\boldsymbol{x}_t) = \boldsymbol{u}_t(\boldsymbol{x}_t) + \frac{b_t(\boldsymbol{x}_t) + \operatorname{div}\boldsymbol{u}_t(\boldsymbol{x}_t) + \frac{1}{2}\varphi^2(t)\left(\Delta \log p_t(\boldsymbol{x}) - \mathcal{R}(\nabla^2 \log p_t(\boldsymbol{x}), \nabla \log p_t(\boldsymbol{x}))\right)}{\|\nabla \log p_t(\boldsymbol{x}_t)\|^2}\nabla \log p_t(\boldsymbol{x}_t), \tag{120}$$

which exactly matches Eq. 15 when $\varphi \equiv 0$ as expected. Now suppose that $\boldsymbol{u}_t = \boldsymbol{u}_t^{\text{PF-ODE}}$ and $b$ is defined as in Eq. 22.

$$b_t^q(\boldsymbol{x}) = -\operatorname{div}\boldsymbol{u}_t(\boldsymbol{x}) - \frac{1}{2}g^2(t)\frac{\sqrt{2D}}{\sigma_t^2}\Phi^{-1}(q). \tag{22}$$

Then the drift becomes

$$\tilde{\boldsymbol{u}}_t(\boldsymbol{x}) = \boldsymbol{u}_t^{\text{DG-ODE}}(\boldsymbol{x}) + \frac{1}{2}\varphi^2(t)\frac{\Delta \log p_t(\boldsymbol{x}) - \mathcal{R}(\nabla^2 \log p_t(\boldsymbol{x}), \nabla \log p_t(\boldsymbol{x}))}{\|\nabla \log p_t(\boldsymbol{x}_t)\|^2}\nabla \log p_t(\boldsymbol{x}_t). \tag{121}$$

**Practical approximation** The Laplacian, $\Delta \log p_t(\boldsymbol{x})$, is given by the trace of the Hessian $\nabla^2 \log p_t(\boldsymbol{x})$, which corresponds to the sum of its eigenvalues. In contrast,

$$\mathcal{R}(\nabla^2 \log p_t(\boldsymbol{x}), \nabla \log p_t(\boldsymbol{x})) = \frac{\nabla \log p_t(\boldsymbol{x})^T \nabla^2 \log p_t(\boldsymbol{x})\nabla \log p_t(\boldsymbol{x})}{\|\nabla \log p_t(\boldsymbol{x})\|^2}$$

represents the Rayleigh quotient of the Hessian evaluated at $\nabla \log p_t(x)$. This quantity is bounded in absolute value by the largest absolute eigenvalue of the Hessian, i.e., its spectral norm. Intuitively, when the eigenvalues of the Hessian are relatively uniform—such as when it is close to a scaled identity matrix—the Laplacian scales linearly with the dimension,

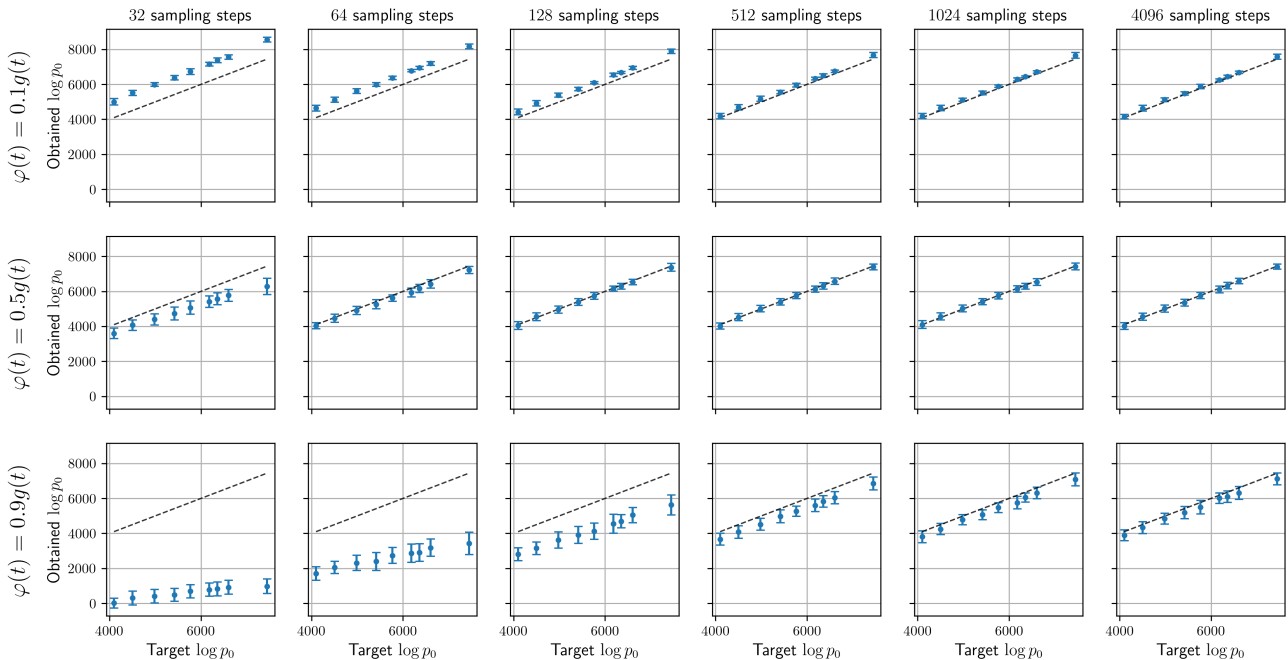

*Figure 15.* **Explicit Quantile Matching obtains exact likelihoods even for stochastic sampling.** Top row: small amount of noise; Middle row: medium amount of noise; Bottom row: large amount of noise. The higher the amount of noise in sampling, the more steps need to be taken for the difference between desired $\log p_0$ and obtained $\log p_0$ to go to zero.

whereas the Rayleigh quotient remains bounded by a constant. This suggests that in high-dimensional settings, the Laplacian dominates.

Empirically, we verified this intuition by estimating both quantities on real data. Specifically, we used a VP-SDE model trained on CIFAR-10 ($32 \times 32$ resolution, $D = 3072$) (Karczewski et al., 2025) and a VE-SDE model trained on ImageNet ($64 \times 64$ resolution, $D = 12288$) (Karras et al., 2022) and sampled uniformly values of $t \in [0, T]$ and corresponding $\boldsymbol{x}_t \sim p_t$ (used 8192 and 16384 samples respectively) and found that the ratio was negligibly small in practice

$$\left| \frac{\mathcal{R}(\nabla^2 \log p_t(\boldsymbol{x}_t), \nabla \log p_t(\boldsymbol{x}_t))}{\Delta \log p_t(\boldsymbol{x}_t)} \right| \approx \begin{cases} 0.0003 \pm 0.00006 & \text{for CIFAR-10} \\ 0.00006 \pm 0.00003 & \text{for ImageNet64} \end{cases} \tag{122}$$

This confirms that, in practice, $\mathcal{R}(\nabla^2 \log p_t(\boldsymbol{x}), \nabla \log p_t(\boldsymbol{x}))$ is negligible compared to $\Delta \log p_t(\boldsymbol{x})$. Therefore, in practice, we use

$$\boldsymbol{u}_t^{\text{DG-SDE}}(\boldsymbol{x}) := \boldsymbol{u}_t^{\text{DG-ODE}}(\boldsymbol{x}) + \underbrace{\frac{1}{2} \varphi^2(t) \frac{\Delta \log p_t(\boldsymbol{x})}{\|\nabla \log p_t(\boldsymbol{x})\|^2} \nabla \log p_t(\boldsymbol{x})}_{\text{correction for added stochasticity}}. \tag{123}$$

## H. Explicit quantile matching with stochastic sampling

In this section, we repeat the experiment from Appendix E, where we define the desired log-density evolution $b_t(\boldsymbol{x}) = \frac{d}{dt}\phi_t$, where $\phi_t$ is the empirical quantile function (see Section 4.1). However, instead of density guidance, we perform stochastic density guidance, i.e. we sample with the density guided SDE:

$$d\boldsymbol{x}_t = \tilde{\boldsymbol{u}}_t(\boldsymbol{x}_t)dt + \varphi(t)\boldsymbol{P}_t(\boldsymbol{x}_t)d\overline{\mathbf{W}}_t, \tag{124}$$

where $\tilde{\boldsymbol{u}}_t$ is defined in Eq. 120, with the exception that we set $\mathcal{R}(\nabla^2 \log p_t(\boldsymbol{x}), \nabla \log p_t(\boldsymbol{x}))$ to zero as explained in Eq. 122. We experimented with different definitions of $\varphi$, which controls the strength of the noise injection, specifically, we tested $\varphi(t) = rg(t)$ for $r = [0.1, 0.5, 0.9]$, where $g$ is the diffusion strength of the forward process Eq. 5. As expected, as the amount of noise increases, the required number of steps to take to achieve exact likelihoods increases. Please see Fig. 15.

# I. JAX implementation of Score Alignment verification

```
1  import jax
2  import jax.random as jr
3  import jax.numpy as jnp
4
5  def aug_drift(u, x, v, t, key):
6      model_key, eps_key = jr.split(key, 2)
7      eps = jr.rademacher(eps_key, (x.size,), dtype=jnp.float32)
8      def u(x_):
9          return u(t, x_.reshape(x.shape), key=model_key).flatten()
10     def du_dv(x_):
11         u_pred, du_dv_pred = jax.jvp(u, (x_,), (v,))
12         return u_pred.reshape(x.shape), du_dv_pred
13     return du_dv(x.flatten())
14
15 def aug_drift_w_omega(u, x, v, t, key):
16     model_key, eps_key = jr.split(key, 2)
17     eps = jr.rademacher(eps_key, (x.size,), dtype=jnp.float32)
18     def u(x_):
19         return u(t, x_.reshape(x.shape), key=model_key).flatten()
20     def du_dv(x_):
21         u_pred, du_dv_pred = jax.jvp(u, (x_,), (v,))
22         return du_dv_pred, u_pred.reshape(x.shape)
23     def div_du_dv(x_):
24         du_dv_pred, du_dv_eps, u_pred = jax.jvp(du_dv, (x_,), (eps,), has_aux=True)
25         return u_pred, du_dv_pred, -jnp.sum(eps * du_dv_eps)
26     return div_du_dv(x.flatten())
27
28 def score_alignment_verification(u, x_T, v_T, T, dt, key, eps=1e-2, use_omega=False):
29     t = T
30     x = x_T
31     v = v_T
32     omega = jnp.sum(v_T **2) if use_omega else None
33     while t > eps:
34         key, subkey = jr.split(key)
35         if use_omega:
36             dx, dv, domega = aug_drift_w_omega(u, x, v, t, subkey)
37             omega -= dt * domega
38         else:
39             dx, dv = aug_drift(u, x, v, t, subkey)
40         x -= dt * dx
41         v -= dt * dv
42         t -= dt
43     if use_omega:
44         return omega
45     else:
46         return jnp.dot(v, score_fn(eps, x, key)) # Assuming score_fn is known
```

*Listing 1.* JAX Implementation of Score Alignment Verification

# J. Quantitative analysis of prior and density guidance

In this section we provide more samples and a quantitative analysis showing that we can reliably control log-density of generated samples and thus amount of detail as measured by PNG file size. Please see: Fig. 16 for results for StableDiffusion, Fig. 17 for EDM2, and Fig. 18 for EDM2 with classifier-free guidance.

# K. More Stochastic Density Guidance samples

We generate more samples using Eq. 25 with the EDM2 model in two scenarios: adding noise early: $\varphi(t) = 0.2g(t)$ for $\log \mathrm{SNR}(t) < -4$ and $\varphi(t) = 0$ otherwise; and adding noise late: $\varphi(t) = 0.3g(t)$ for $\log \mathrm{SNR}(t) > -3$ and $\varphi(t) = 0$ otherwise. To demonstrate that we have fine-grained control over image detail, we do it for various values of the

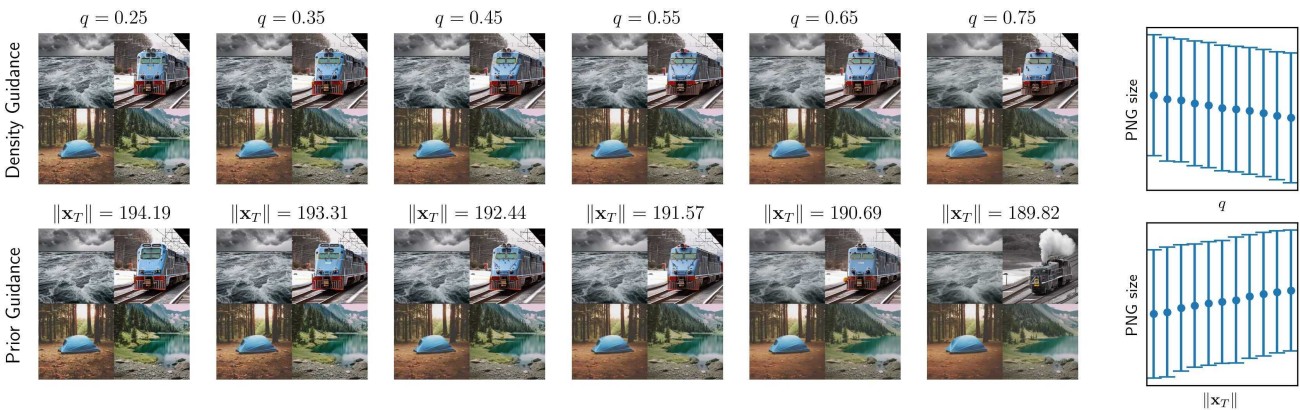

*Figure 16.* **Stable Diffusion v2.1 samples.** PG and DG can monotonically control the amount of detail as measured by PNG file size.

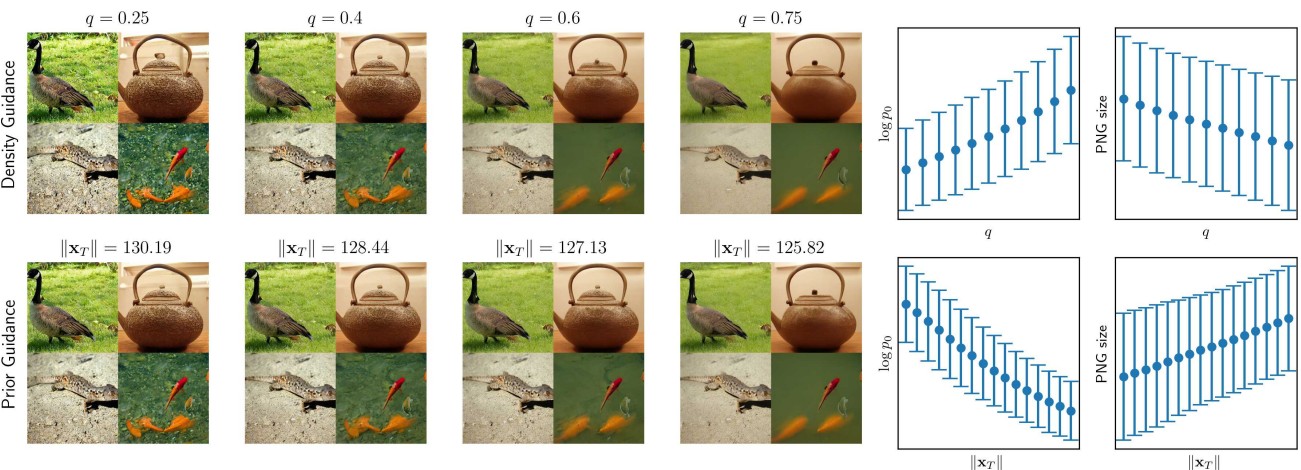

*Figure 17.* **EDM2 samples.** PG and DG monotonically control log-density and amount of detail.

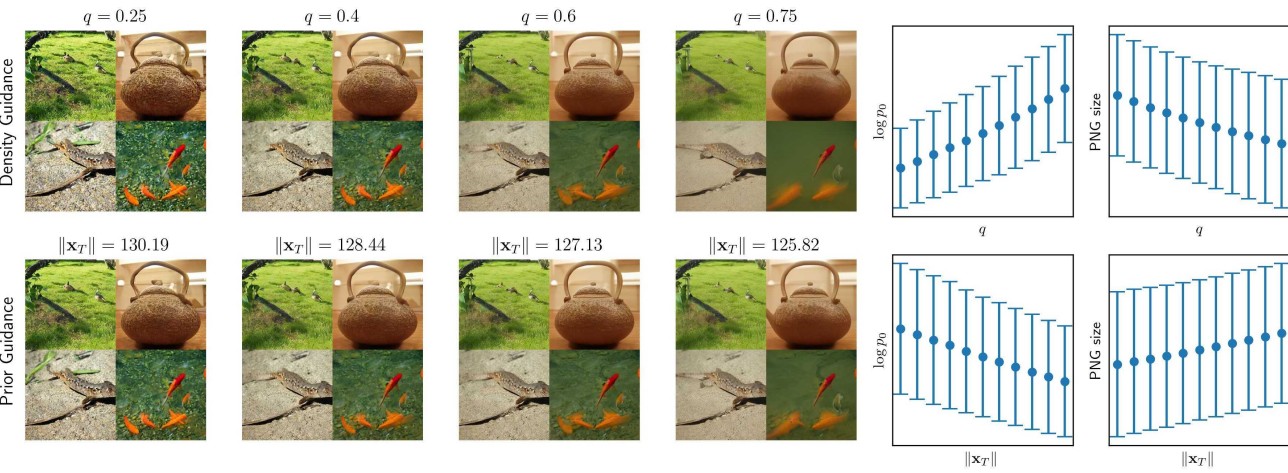

*Figure 18.* **EDM2 samples with classifier-free guidance.** Both PG and DG are effective when CFG is used.

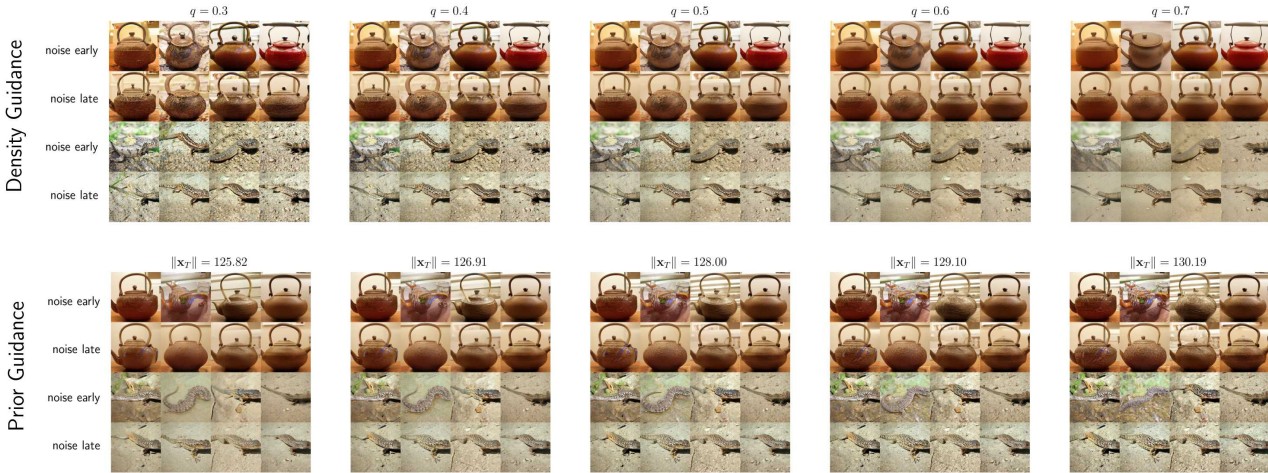

*Figure 19.* **Density guided samples in stochastic sampling.** We chose two starting random seeds and for each of them generated 4 random samples with two strategies: adding noise early in the generation (altering high-level detail), or late (low-level detail). For the inner sampling loop, we used the same random seeds across all runs.

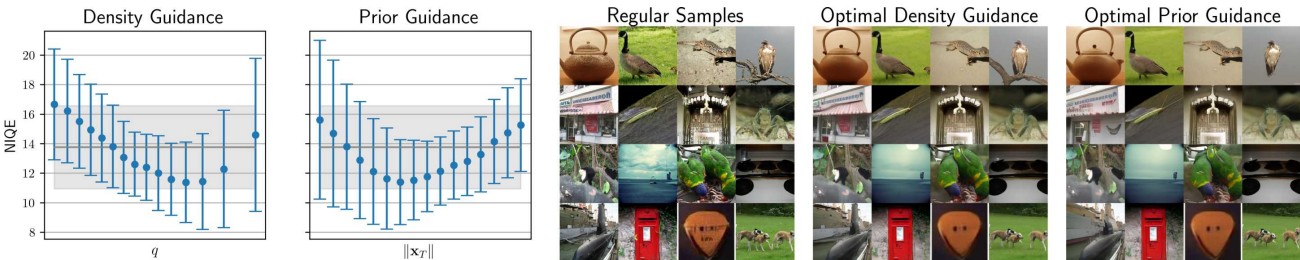

*Figure 20.* **Density and Prior Guidance can improve perceptual quality as measured by NIQE score.** Left: NIQE scores (lower is better) for both Density and Prior Guidance on the EDM2 model for various values of the hyperparameters ($q$ and $\|\boldsymbol{x}_T\|$ respectively) with regular samples as reference in grey. Right: Representative examples of best-scoring hyperparameters. NIQE score seems to favor lower-detail images.

hyperparameters. See Fig. 19 for a visualization. We compare this for Prior Guidance, which is not principled for stochastic sampling, because the larger the amount of noise, the less information $\boldsymbol{x}_T$ carries about the final generated sample $\boldsymbol{x}_0$.

## L. Connection with perceptual metrics

There exist various metrics measuring the perceptual quality of image generation models. Most popular include LPIPS (Zhang et al., 2018) and SSIM (Wang et al., 2004). However, these are *reference-based* quality measures, meaning that they require the ground truth to compare to, which is not available in unconditional image generation. We therefore used NIQE (Mittal et al., 2012), which is a non-reference perceptual quality measure, reported to strongly correlate with human judgement. It provides a single number per image, which indicates whether an image has been distorted (a lower number - higher quality). It was used, e.g., by Sami et al. (2024) to evaluate super-resolution diffusion models.

## M. Prior Guidance samples

In Fig. 21 we show samples generated with the EDM2 ImageNet $512 \times 512$ model (Karras et al., 2024b). Specifically, we randomly sampled a latent code $\boldsymbol{x}_T = \sigma_T \boldsymbol{\varepsilon}$ for $\boldsymbol{\varepsilon} \sim \mathcal{N}(\boldsymbol{0}, \boldsymbol{I}_D)$ and scaled it to have specific values of the norm. Since $\log p_T(\boldsymbol{x}_T) = C - \frac{1}{2}\|\boldsymbol{\varepsilon}\|^2$ and $\|\boldsymbol{\varepsilon}\|^2 \sim \chi^2(D)$, we choose the values of the target squared norm to be quantiles of $\chi^2(D)$ for $q \in [0.001, 0.999]$ for $\boldsymbol{x}_T$ to remain in the typical region of $p_T$. Higher values of $q$ mean higher norm, i.e. lower $\log p_T(\boldsymbol{x}_T)$, and are thus decoding produces more detailed images.

Given that the Score Alignment holds for the EDM2 model (Fig. 3), we can see that scaling the latent code (Prior Guidance) is effective in controlling $\log p_0(\boldsymbol{x}_0)$ and thus the image detail. We additionally include samples for StableDiffusion v2.1 (Rombach et al., 2022) in Fig. 22.

## N. Gaussian Mixture asymptotics

**Lemma N.1.** *Let $f : \mathbb{R} \to \mathbb{R}$, $f(x) = \exp(-\frac{1}{2}x)x$. Then for all $x \in \mathbb{R}$*

$$f(x) \leq 1 \tag{125}$$

*Proof.*

$$f'(x) = -\frac{1}{2}\exp\left(-\frac{1}{2}x\right)x + \exp\left(-\frac{1}{2}x\right) = \exp\left(-\frac{1}{2}x\right)\left(1 - \frac{1}{2}x\right).$$

$f'(x) > 0$ for $x < 2$ and $f'(x) < 0$ for $x > 2$. Therefore for all $x \in \mathbb{R}$

$$f(x) \leq f(2) = \exp(-1)2 = \frac{2}{e} < 1.$$

$\square$

**Lemma N.2.** *For any $a > 0$ and $x > 0$*

$$\left(\frac{a}{x} + x\right)^2 \geq 4a. \tag{126}$$

*Proof.*

$$0 \leq \left(\frac{a}{x} - x\right)^2 = \frac{a^2}{x^2} - 2a + x^2 = \left(\frac{a}{x} + x\right)^2 - 4a$$

$\square$

**Theorem 1.** *Let $X \in \mathbb{R}^D$ be drawn from a mixture of $K$ Gaussian components:*

$$\Pr(Y = k) = \pi_k, \quad X \mid (Y = k) \sim \mathcal{N}(\mu_k, \sigma^2 I_D), \tag{127}$$

*where $\mu_j \neq \mu_i$ for $i \neq j$ (note that this assumption is not restrictive, because when different components share the mean, the mixture can be rewritten with distinct $\mu$ and updated $\pi$). Define*

$$h(X) = \frac{1}{\sqrt{2D}}\left[\sum_{j=1}^{K} w_j(X)\frac{\|X - \mu_j\|^2}{\sigma^2} - D\right], \tag{128}$$

*where*

$$w_j(x) = \frac{\pi_j \exp\left[-\frac{1}{2\sigma^2}\|x - \mu_j\|^2\right]}{\sum_{m=1}^{K} \pi_m \exp\left[-\frac{1}{2\sigma^2}\|x - \mu_m\|^2\right]}. \tag{129}$$

*Then $h(X) \xrightarrow{d} N(0,1)$ as $D \to \infty$.*

*Proof.* **Step 1: It suffices to show $h(X) \mid (Y = k) \to N(0,1)$.** Indeed,

$$\varphi_{h(X)}(t) = \mathbb{E}\left[e^{ith(X)}\right] = \sum_{k=1}^{K} \pi_k \mathbb{E}\left[e^{ith(X)} \mid Y = k\right] = \sum_{k=1}^{K} \pi_k \varphi_{h(X)|Y=k}(t).$$

Thus if each conditional law converges to $N(0,1)$ (point-wise convergence of characteristic functions), so does the unconditional mixture.

**Step 2: Rewrite $X = \mu_k + \sigma\varepsilon$.** Conditioning on $Y = k$, we have $X = \mu_k + \sigma\varepsilon$, $\varepsilon \sim \mathcal{N}(\mathbf{0}, \boldsymbol{I}_D)$. Thus

$$\frac{\|X - \mu_k\|^2}{\sigma^2} = \|\varepsilon\|^2 \sim \chi_D^2, \quad \frac{\|\varepsilon\|^2 - D}{\sqrt{2D}} \xrightarrow{d} N(0,1)$$

**Step 3: Define remainder $\mathcal{R}_D$.** Set

$$\sum_{j=1}^{K} w_j(X) \frac{\|X - \mu_j\|^2}{\sigma^2} = \|\varepsilon\|^2 + \mathcal{R}_D,$$

so

$$h(X) = \frac{\|\varepsilon\|^2 - D}{\sqrt{2D}} + \frac{\mathcal{R}_D}{\sqrt{2D}}.$$

Using Slutsky's theorem, it suffices to show $\mathcal{R}_D / \sqrt{2D} \xrightarrow{d} 0$, which is a weaker condition than convergence in probability. **It thus suffices to show**

$$\frac{\mathcal{R}_D}{\sqrt{2D}} \xrightarrow{\mathbb{P}} 0 \tag{130}$$

**Step 4: Showing $\mathcal{R}_D / \sqrt{2D} \xrightarrow{\mathbb{P}} 0$.** Let

$$\Delta_{j,k} := \frac{\mu_k - \mu_j}{\sigma} \neq \mathbf{0}.$$

Note

$$\frac{\|X - \mu_j\|^2}{\sigma^2} = \|\varepsilon + \Delta_{j,k}\|^2 = \|\varepsilon\|^2 + 2\varepsilon^T \Delta_{j,k} + \|\Delta_{j,k}\|^2 = \|\varepsilon\|^2 + b_j$$

for $b_j = 2\varepsilon^T \Delta_{j,k} + \|\Delta_{j,k}\|^2$ and $b_k = 0$ since $\Delta_{k,k} = 0$. Hence

$$\mathcal{R}_D = \sum_{j=1}^{K} w_j(X) \left[2\varepsilon^T \Delta_{j,k} + \|\Delta_{j,k}\|^2\right] = \sum_{j \neq k} w_j(X) b_j.$$

Also:

$$w_j(X) = \frac{\pi_j \exp\left[-\frac{1}{2\sigma^2}\|x - \mu_j\|^2\right]}{\sum_{m=1}^{K} \pi_m \exp\left[-\frac{1}{2\sigma^2}\|x - \mu_m\|^2\right]} = \frac{\pi_j \exp\left[-\frac{1}{2}(\|\varepsilon\|^2 + b_j)\right]}{\sum_{m=1}^{K} \pi_m \exp\left[-\frac{1}{2}(\|\varepsilon\|^2 + b_m)\right]} = \frac{\pi_j \exp(-\frac{1}{2}b_j)}{\sum_{m=1}^{K} \pi_m \exp(-\frac{1}{2}b_m)}.$$

Then

$$\mathcal{R}_D = \frac{\sum_{j \neq k} \pi_j \exp(-\frac{1}{2}b_j) b_j}{\pi_k + \sum_{m \neq k} \pi_m \exp(-\frac{1}{2}b_m)}.$$

We will now separate the sum in the numerator of $\mathcal{R}_D$ into positive and negative $b_j$. Define $J^+ = \{j \neq k | b_j \geq 0\}$ and $J^- = \{j \neq k | b_j < 0\}$:

$$\mathcal{R}_D = \frac{\sum_{j \in J^+} \pi_j \exp(-\frac{1}{2}b_j) b_j}{\pi_k + \sum_{m \neq k} \pi_m \exp(-\frac{1}{2}b_m)} + \frac{\sum_{j \in J^-} \pi_j \exp(-\frac{1}{2}b_j) b_j}{\pi_k + \sum_{m \neq k} \pi_m \exp(-\frac{1}{2}b_m)}.$$

From Lemma N.1, $\exp(-\frac{1}{2}b_j) b_j \leq 1$ and thus $\sum_{j \in J^+} \pi_j \exp(-\frac{1}{2}b_j) b_j \leq 1$. Therefore (note shrinking denominators to achieve upper bounds):

$$|\mathcal{R}_D| \leq \left| \frac{\sum_{j \in J^+} \pi_j \exp(-\frac{1}{2}b_j) b_j}{\pi_k + \sum_{m \neq k} \pi_m \exp(-\frac{1}{2}b_m)} \right| + \left| \frac{\sum_{j \in J^-} \pi_j \exp(-\frac{1}{2}b_j) b_j}{\pi_k + \sum_{m \neq k} \pi_m \exp(-\frac{1}{2}b_m)} \right|$$

$$\leq \frac{1}{\pi_k + \sum_{m \neq k} \pi_m \exp(-\frac{1}{2}b_m)} + \frac{\sum_{j \in J^-} \pi_j \exp(-\frac{1}{2}b_j) |b_j|}{\pi_k + \sum_{m \neq k} \pi_m \exp(-\frac{1}{2}b_m)}$$

$$\leq \frac{1}{\pi_k} + \frac{\sum_{j \in J^-} \pi_j \exp(-\frac{1}{2}b_j) |b_j|}{\pi_k + \sum_{m \in J^-} \pi_m \exp(-\frac{1}{2}b_m)}$$

$$\leq \frac{1}{\pi_k} + \frac{\sum_{j \in J^-} \pi_j \exp(-\frac{1}{2}b_j) |b_j|}{\pi_k + \sum_{m \in J^-} \pi_m \exp(-\frac{1}{2}b_m)} = \frac{1}{\pi_k} + \sum_{j \in J^-} \tilde{w}_j |b_j|,$$

where $\tilde{w}_j = \frac{\pi_j \exp(-\frac{1}{2}b_j)}{\pi_k + \sum_{m \in J^-} \pi_m \exp(-\frac{1}{2}b_m)}$ and $\sum_{j \in J^-} \tilde{w}_j = \frac{\sum_{j \in J^-} \pi_j \exp(-\frac{1}{2}b_j)}{\pi_k + \sum_{m \in J^-} \pi_m \exp(-\frac{1}{2}b_m)} \leq 1$. Thus

$$\sum_{j \in J^-} \tilde{w}_j |b_j| \leq \sum_{j \in J^-} \tilde{w}_j \max_{j \in J^-} |b_j| \leq \max_{j \in J^-} |b_j| = \max(-\min_{j \neq k} b_j, 0),$$

where the last equality comes from the definition of $J^-$. In summary

$$|\mathcal{R}_D| \leq \frac{1}{\pi_k} + \max(-\min_{j \neq k} b_j, 0).$$

Now, for any $\delta > 0$:

$$\Pr\left(\frac{|\mathcal{R}_D|}{\sqrt{2D}} > \delta\right) = \Pr\left(|\mathcal{R}_D| > \delta\sqrt{2D}\right) \leq \Pr\left(\max(-\min_{j \neq k} b_j, 0) > \delta\sqrt{2D} - \frac{1}{\pi_k}\right).$$

Now choose $D$ large enough so that $a = \delta\sqrt{2D} - \frac{1}{\pi_k} > 0$. Then

$$\Pr\left(\max(-\min_{j \neq k} b_j, 0) > a\right) = \Pr\left(\exists_{j \neq k}\, b_j < -a\right) \leq \sum_{j \neq k} \Pr\left(b_j < -a\right).$$

Now plugging in the definition of $b_j$ and using the fact that $Z = \frac{\varepsilon^T \Delta_{k,j}}{\|\Delta_{k,j}\|} \sim \mathcal{N}(0,1)$.

$$\Pr\left(b_j < -a\right) = \Pr\left(2\varepsilon^T \Delta_{j,k} + \|\Delta_{j,k}\|^2 < -a\right) = \Pr\left(Z < -\frac{1}{2}\left(\frac{a}{\|\Delta_{j,k}\|} + \|\Delta_{j,k}\|\right)\right).$$

And using the standard Gaussian tail bound $\Pr(Z < -t) = \Pr(Z > t) \leq \exp(-\frac{1}{2}t^2)$ for $t > 0$:

$$\Pr\left(b_j < -a\right) \leq \exp\left(-\frac{1}{8}\left(\frac{a}{\|\Delta_{j,k}\|} + \|\Delta_{j,k}\|\right)^2\right) \leq \exp\left(-\frac{1}{2}a\right),$$

where the last inequality comes from Lemma N.2. Therefore

$$\Pr\left(\frac{|\mathcal{R}_D|}{\sqrt{2D}} > \delta\right) \leq \sum_{j \neq k} \Pr(b_j < -a) \leq \sum_{j \neq k} \exp\left(-\frac{1}{2}a\right) = (K-1)\exp\left(-\frac{1}{2}\left(\delta\sqrt{2D} - \frac{1}{\pi_k}\right)\right) \xrightarrow[D \to \infty]{} 0$$

We have shown that for all $\delta > 0$

$$\lim_{D \to \infty} \Pr\left(\left|\frac{\mathcal{R}_D}{\sqrt{2D}}\right| > \delta\right) = 0, \tag{131}$$

which means

$$\frac{\mathcal{R}_D}{\sqrt{2D}} \xrightarrow[D \to \infty]{\mathbb{P}} 0.$$

$\square$

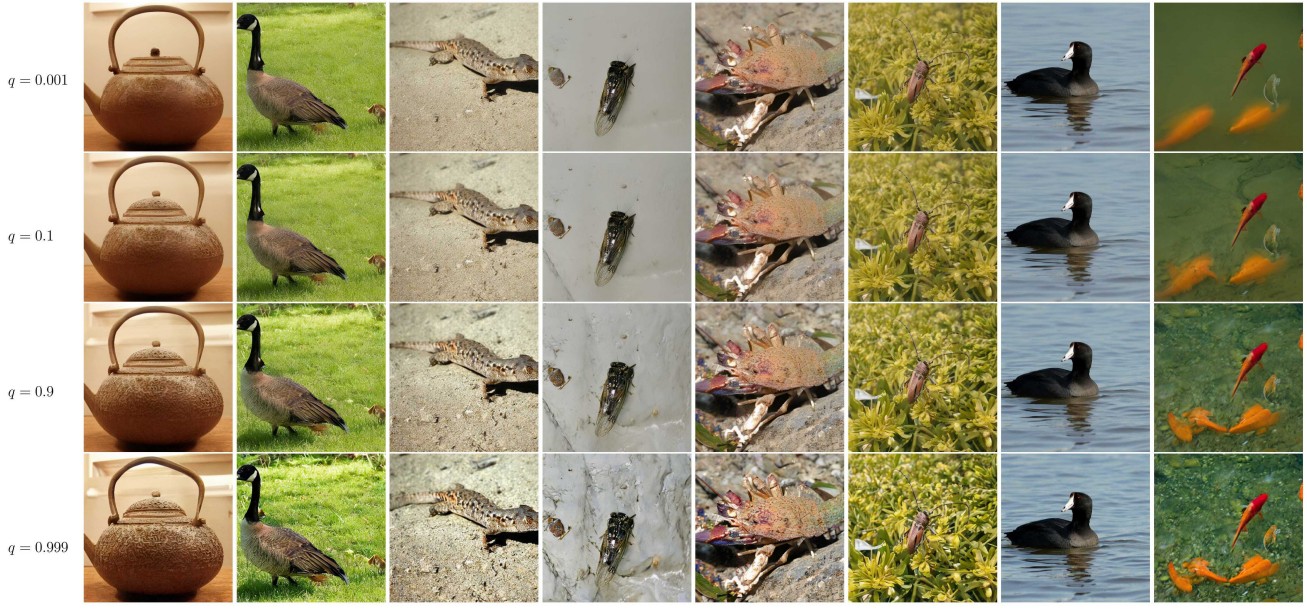

*Figure 21.* **Prior Guidance controls the detail when Score Alignment holds.** Samples generated with the EDM2 latent diffusion model (Karras et al., 2024b) for different values of quantiles $q$ for the $\chi^2$ distribution. See Appendix M for details.

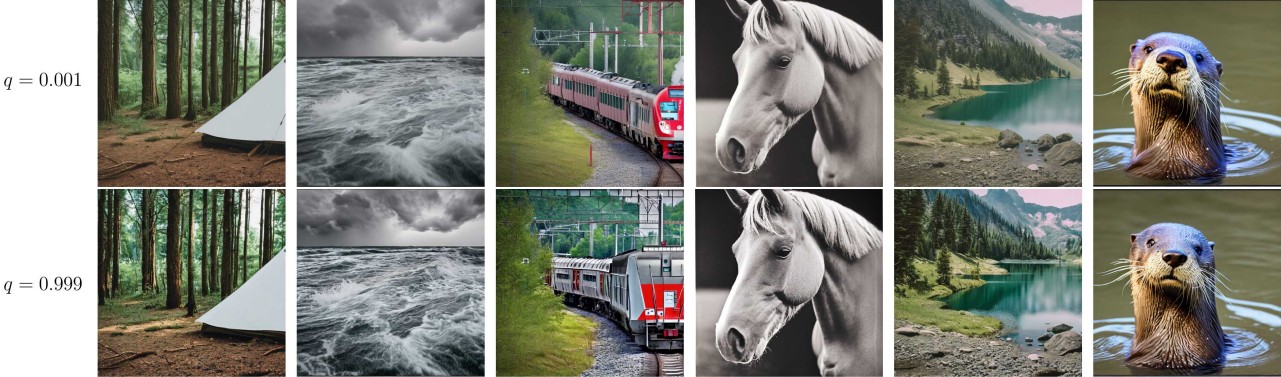

*Figure 22.* Samples with Prior Guidance on StableDiffusion v2.1 (Rombach et al., 2022)

