# OpenReview forum: "Devil is in the Details: Density Guidance for Detail-Aware Generation with Flow Models"
_ICML.cc/2025/Conference — ICML 2025 poster_

### Official Review · Reviewer_ppbq · 2025-03-14

**Overall Recommendation:** 3

**Summary:**

This paper introduces a collection of methods for controlling likelihood of samples generated by a flow/diffusion model. Authors provide a comprehensive review of prior work on density control, in particular providing a more formal analysis of latent scaling [Song 2021]. They further introduce density guidance - a method for sampling with explicit likelihood control through an alternative ODE formulation ensuring the sample stays in a pre-defined quantile over time. They further introduce a stochastic variant of density guidance.

**Claims And Evidence:**

The paper does deliver on the theoretical claims, but the experimental claims are not very extensively validated (e.g. empirical evaluation of prior vs density vs stochastic density guidance are a handful of qualitative examples).

**Essential References Not Discussed:**

N/A

**Experimental Designs Or Analyses:**

N/A

**Methods And Evaluation Criteria:**

There is almost no coherent evaluation.

**Other Comments Or Suggestions:**

Although this is mostly theory-focused work, it will be beneficial to get minimal quant/qual validation, even if on toy data.

**Other Strengths And Weaknesses:**

+ Overall the paper introduces a significant contribution to understanding the properties of generative diffusion models, both in terms of providing theoretical insight to existing sampling techniques, and in terms of novel methods for density control during generation.
- The practical utility of proposed method is a bit unclear due to lack of evaluation.

**Questions For Authors:**

Apart from several images, experimental validation is missing, would be great to understand whether the theoretical claims could be properly validated? E.g. why not follow methodology from [Karczewski'25]  for quantiative/qualitative analysis?

**Relation To Broader Scientific Literature:**

This paper explores an interesting property of the flow/diffusion models that has been noticed recently [Karczewski'25] and explains ad-hoc techniques commonly used in score-based generative models [Song'25].

**Theoretical Claims:**

I checked the claims in the main paper leading to Eq.(24), assuming that proves in supp are correct, they do seem to be meaningful and consistent with existing work [Karczewski'ICLR2025]

---

> ### Author Rebuttal · Authors · 2025-04-01
>
> We would like to thank the Reviewer for their time and efforts to scrutinize our submission. We address the raised concern below.
>
> **File with new figures:** https://anonymous.4open.science/r/DensityGuidance-20E6/Density_guided_sampling___Rebuttal.pdf
>
> >The practical utility of the proposed method is a bit unclear due to lack of evaluation. It would be great to understand whether the theoretical claims could be properly validated? E.g. why not follow methodology from [Karczewski'25] for quantiative/qualitative analysis?
>
> Thank you for this suggestion, we have now included an extensive evaluation of the proposed methods building on the methodology of [Karczewski’25]. Specifically:
>
> ### Explicit Quantile Matching (EQM)
>
> We estimated the quantile function for the CIFAR model as described in line (311 left). We tested $K = [16, 32, 64, 128, 256, 512, 1024]$ and found that using $K=128$ is enough to ensure a correlation between the desired value of log-density and the obtained one is above 99%. Based on the estimated quantile function $\phi_t$, we estimate $b_t = \frac{d}{dt}\phi_t$ with a moving average of the finite difference estimates.
>
> Furthermore, we found that the difference between the desired values of log-density and the obtained ones goes to zero as we decrease the discretization error (increase the number of sampling steps). Interestingly, for lower number of sampling steps, even though we do not obtain exact desired values of likelihood, the correlation between the desired values and the obtained ones remains above 99%, even for as few as 32 Euler sampling steps. This means that for all values of the number of sampling steps, we saw a monotonic relationship between the target $\log p_0$ and the amount of detail (PNG size). Please see Figure 17.
>
> Finally, we also show that **we can obtain exact values of likelihoods even when sampling stochastically** by using results from Appendix F, and the Euler–Maruyama algorithm. We tested different amounts of added noise: $\varphi(t)= r g(t)$ for $r=[0.1, 0.5, 0.9]$. As expected, as the amount of noise increases, the required number of steps to take to achieve exact likelihoods also increases. Please see Figure 18.
>
> ### Prior Guidance vs Density Guidance vs Stochastic Density Guidance
>
> We quantitatively compared Density Guidance (DG), and Prior Guidance (PG) using the EDM2 model. We measures the correlation between the hyperparameter ($q$ for Density guidance and $||x_T||$ for prior guidance) and the obtained $\log p_0$. We found 66% for DG and 68% for PG.
>
> Furthermore, we compared DG and PG with stochastic sampling. I.e.using Eq 25 for Stochastic Density Guidance (SDG), and Eq 7 for “Stochastic Prior Guidance” SPG (i.e. regular stochastic sampling after rescaling the latent code). We tested two scenarios:
> Adding noise early: $\varphi(t)=0.2g(t)$ for $\log SNR(t) < -4.03$, and $\varphi(t)=0$ otherwise;
> Adding noise late $\varphi(t)=0.3g(t)$ for $\log SNR(t) > -3$, and $\varphi(t)=0$ otherwise.
>
> We found the correlation between the hyperparameter and the obtained $\log p_0$ to be 50% for SDG and 25% for SPG. We summarize all correlations in the table below.
>
> ||Density Guidance|Prior Guidance|
> |-|-|-|
> |Deterministic Sampling|66%|68%|
> |Stochastic Sampling|50%|25%|
>
> For DG the drop in correlation from deterministic to stochastic sampling can be explained by the same reasoning as for the EQM, i.e. stochastic sampling requires significantly more sampling steps to achieve the desired levels of $\log p_0$ (Figure 18).
>
> For PG, stochastic sampling is not principled, i.e. the more noise we add during sampling, the less information is contained in the starting point $x_T$. For example, if $\varphi(t)=g(t)$ for all $t$, then the process is the Reverse SDE, and $p(x_0|x_T)$ does not depend on $x_T$, and thus scaling the latent code has no effect on the final sample. Hence the need for Density Guidance for stochastic sampling.
>
> Please see Fig 20 for details on the evaluation of log-densities and corresponding PNG files sizes, and Fig 21 for the visualization of the stochastic samples.
>
> ### Additional Experiments
>
> We have also added the following:
> * Analysis of the impact of Guidance on perceptual metric NIQE (Fig 14, more details in the response to Reviewer 8ij3)
> * More samples and quantitative results with Stable Diffusion (Fig 19)
> * Results with models using Classifier-Free Guidance (Fig 22, more details in the response to Reviewer reGE, Q9)
> * A new State-of-the-art model FLUX (Fig 23-25, more details in the response to Reviewer reGE, under SOTA models)
> * A rigorous proof of the hypothesis we posed in Appendix D about the asymptotic behaviour of $h(x)$ for the Gaussian Mixture (Theorem 1 in the uploaded file)
>
> We thank the Reviewer again for their constructive feedback, which strengthened our claims and improved the quality of our submission. We hope that we have adequately addressed the concerns, and you will consider raising your score.

---

### Official Review · Reviewer_reGE · 2025-03-14

**Overall Recommendation:** 4

**Summary:**

This paper studies the control of the amount of details in samples from diffusion models. The authors first established a theoretical framework (Score alignment) to explain a trick to increase sample details in prior literature. Then, the authors explore a suite of methods that can be used to control the exact scale of amount of details in the generated samples, both in the deterministic case (Density guidance) and in stochastic case. Theoretical claims are proved on simplified cases and some qualititave anlaysis is performed. Experiments on SD2.1 and EDM are performed to show the real-world use cases.

**Claims And Evidence:**

**Major claims made in Sec. 3**:

- The trick of "scaling latent code" will work because (1) it decreases the likelihood of $x_T$, (2) decreasing likelihood of $x_T$ correlates to the decrease of likelihood of $x_0$, and (3) the likelihood of $x_0$ correlates to the amount of details. (1) is supported by the prior Gaussian distribution. (2) is supported by the Score Alignment condition, which is partially and qualitatively shown for selected models. (3) is supported by a previous literature.
  - **(Q1)** The major complaint here is on (2). Only two models (VP-SDE and EDM2) are analyzed qualitatively. It would be great if the authors could analyze state-of-the-art models (e.g., Stable diffusion XL, FLUX). Moreover there is no theoretical guarantee for this property.
- Likelihood correlates well with amount of details in the image, which is shown through the correlation between the image compression size and the sample likelihood.

**Major claims made in Sec. 4**

- Explicit quantile matching enables sampling images with an exact likelihood of $c$. This is mostly supported with theoretcial proof in Appendix D,  and a claim from prior literature that sampling from typical regions will produce accurate predictions.
  - **(Q2)** The major complaint here is that this claim is not supported empirically. Practically, the authors propose to sample $K$ times to estimate the quantile function. However it is not clear on how large $K$ should be in practice. Moreover, there is no empirical results on whether using this method indeed produce samples with a likelihood of the exact value. It would be great if we can see results of this algorithm to be applied to real-world unconditional generators, and whether the produced samples indeed have an altered amount of detail and whether those samples indeed have an exact likelihood expected.
- Implicit quantile matching works similarly for conditional generators. This is supported by some sparse theoretical results in Appendix D. and qualitative results in Fig. 9.
  - **(Q3)** I could not fully understand why using $b_t$ as defined in Eq. 21 would guarantee the condition in Eq. 18. I can get some intuitions from the fact in Eq. 20, yet is there a rigorous proof on this?
  - **(Q4)** The results in Fig. 9 are interesting. However, the result for stable diffusion is sparse: there are only two levels shown, so we could not really tell whether the method is really achieving a fine-grained control over amount of detail. Moreover in SD2.1 it can be seen that there are changes in semantic contents, especially in the train example.
  - **(Q5)** There are no quantitative analysis on the generated samples' amount of details. It would be beneficial if we can see a plot or a table measuring both the image compression size (in PNG as in Fig.4) and the likelihood of $K$ samples in different levels.

**Major claims made in Sec. 5**

- Eq 25. extends the above sampling procedure to stochastic process. This is supported by the proof in Appendix F and by experiments in Fig. 10.
  - **(Q6)** There are only two levels of details in Fig. 10. Similarly, we could not tell whether the proposed method works to control specific level or details. (Will a simple prior guidance do very similar thing?)

**Essential References Not Discussed:**

Relevant literatures are adequately discussed to the best of the reviewer's knowledge.

**Experimental Designs Or Analyses:**

Yes, I checked most experimental designs and analyses. They are mostly adequate except for some minor issues as discussed in Q2, Q4, Q5 and Q6.

**Methods And Evaluation Criteria:**

The methods proposed in this paper are sound from the description. The evaluation mostly makes sense except for some minor issues, as discussed in Q2, Q4, Q5, and Q6.

**Other Comments Or Suggestions:**

Minor issues and comments:

- In L175, it seems that "$v_t$" is not discussed in Eq. 11.
- There should be reference for Eq. 3.

**Other Strengths And Weaknesses:**

Beyond the strengths and weaknesses already discussed (Q1-Q6):

Strengths:

- The paper is well-written and well-organized. Detailed discussions and proofs are presented for most claims.
- Extensive studies are conducted for the research question proposed. The contribution is solid and extensive, and the paper explores the proposed framework in various settings including conditional/unconditional generation and stochastic generation.
- The problem studied in this paper is interesting. It may potentially be insightful to inspire reseach in related fields.

Weaknesses:

- There are some issues with the validation and experiment design, as in Q1-Q6.
- **(Q7)** What would be a practical application scenario of the proposed technique? When would a user be interested in a fine-detailed control over the amount of details of the generated image?
- **(Q8)** There is no discussion on when the method will fails to achieve the desired properties. There are some approximations in the theoretical proofs of the method, so it would be great if the authors could analyze the scenarios where the method will fail.
- **(Q9)** How would the method be used together with Classifier-free guidance, which is the de facto methods to perform conditional generation for diffusion models?

**Questions For Authors:**

Minor questions:

- L140(right), I actually didn't find such claims explicitly stated in Song et al. 2021b. Could you point out the exact location of such a claim?
- Will the proposed method be able to work for state-of-the-art models like Stable Diffusion XL or FLUX?

**Relation To Broader Scientific Literature:**

This paper studies models from recent studies in diffusion models (Song et al. 2021) and flow models (Lipman et al. 2023, Liu et al. 2023) and leverages some insights from prior literature (Song et al. 2021b). This paper is strongly based on the findings in (Karczewski el al. 2024).

**Theoretical Claims:**

I didn't fully check the proofs in the appendix.

---

> ### Author Rebuttal · Authors · 2025-04-01
>
> We thank the Reviewer for their thorough evaluation of our work and very insightful questions! We address the raised points below.
>
> **File with new figures:** https://anonymous.4open.science/r/DensityGuidance-20E6/Density_guided_sampling___Rebuttal.pdf
>
> Glossary:
> * Density Guidance - DG
> * Prior Guidance - PG
> * Score Alignment - SA
> * Classifier-free guidance - CFG
>
> > **Q1:** Only two models analyzed for SA, and no theoretical guarantee.
>
> This is true that there is no theoretical guarantee for SA. There cannot be such a guarantee as we show:
>   * In Fig 3, right - for the CIFAR model, SA does not hold for 3% of the latent codes (line 209)
>   * In Appendix C.3, we provide an example of a Gaussian mixture (exact scores known), where SA does not hold.
>
> This emphasizes the point from the paper: **SA does not always hold.** This in part motivates the DG approach, because PG is not always guaranteed to work.
>
> We discuss FLUX at the end of the response.
>
> > **Q2:** no quantitative evidence of Explicit Quantile Matching.
>
> Please see our response to Reviewer ppbq.
>
> > **Q3:** Can you prove that Eq 21 implies Eq 18?
>
> We actually do not make that claim. Eq 21 is based on results in Appendix D, which **we have now extended by a rigorous proof in the Gaussian Mixture case** (Theorem 1). The motivation is to keep the samples in the typical regions of $p_t$, but we do not guarantee exact quantiles.
>
> > **Q4:** Few StableDiffusion samples. Also, semantic changes visible.
>
> We have included more levels for Stable Diffusion, with PNG sizes (Fig 19). Regarding semantic changes - this is true, and can be even more drastic as in the train example with PG in Fig 19. We do not guarantee that only the low-level features change in all cases. In the extremes semantic changes can happen as well. However, it is consistent with the amount of detail as measured by PNG.
>
> > **Q5:** Quantitative analysis of DG
>
> Please see our response to Reviewer ppbq.
>
> > **Q6:** Only two levels in Fig 10. Does stochastic guidance differ from PG?
>
> We added more samples and levels in Fig 21 for both DG and PG. Perceptually, both seem to be monotonically controlling the detail, but we argue in the response to Q5: DG is more accurate.
>
> > **Q7:** What are practical application scenarios?
>
> Due to character limit, please refer to our response to Reviewer J9Xk, where we discuss potential applications.
>
> > **Q8:** When might the method fail?
>
> A potential issue is applying DG in low dimensions. We use the fact that $h(x)$ is approximately Gaussian (Appendix D). This only holds when the dimensionality is large.
>
> Another approximation we make is discussed in lines 1147-1161. We justify it on two datasets. If one wants to increase the accuracy further, Eq 119 can be used instead, which makes no approximations. It comes at a cost of one additional Jacobian-Vector-Product.
>
> > **Q9:** Would the method work with CFG?
>
> As explained in the CFG paper, CFG can be interpreted as classifier guidance with an implicit classifier. This means that CFG is just a regular diffusion process, but with a different base distribution, favouring a certain class. Thus, all our results apply without changes, just with a redefined target distribution.
>
> We sampled with an EDM2 model with CFG and found consistent behaviour with other models. Please see Fig 22.
>
> >L175, "$v_t$" is not discussed
>
> We denote by $v_t$ the score pushed forward from $T$ to $t$. We refer to it later in the text, e.g. in Eq 12. We will make it more explicit in the final revision.
>
> > No reference for Eq. 3.
>
> The reference is Chen et al. 2018. We will make it more explicit.
>
> >I didn't find such claims in Song et al. 2021b
>
> It can be seen in the Appendix in Figure 6 - it is not referenced in the main text. Authors call it "temperature rescaling" (reducing norm of embedding).
>
> > Will the methods work for SOTA models like FLUX?
>
> We have included new results with FLUX.1[dev]. Fig 23 shows samples with PG and DG, and Fig 24 shows a DG image with PNG and TIFF filesize comparison. Fig 25 shows the coupling of PNG and TIFF filesizes over guidance.
>
> FLUX shows different behavior to other models. Images are richer, but detail variations from guidance are milder. The weaker DG effect can be attributed to the FLUX model coupling a latent diffusion on 16x64x64 space with a strong decoder to 3x768x768, whose effect on logp is unknown. We only control the latent portion of the model. FLUX is undocumented and unpublished.
>
> The effect of DG on PNG filesizes also becomes inconsistent: adding more semantic detail doesn’t necessarily increase filesize, possibly due to the images already being highly realistic and rich in patterns. Furthermore, PNG is only an approximation of the true information content of the image. We include comparison to TIFFs, which shows more consistent coupling between filesize and detail.
>
> We thank the Reviewer again for their high-quality review that significantly contributed to improving our submission.

---

> > ### Comment · Reviewer_reGE · 2025-04-04
> >
> > Thanks for the rebuttal! My initial concerns have been addressed.

---

> > > ### Author Response · Authors · 2025-04-05
> > >
> > > We would like to thank the reviewer again for their high-quality and thorough review, as well as the constructive feedback provided. We are glad to hear that the concerns have been addressed to the reviewer's satisfaction and appreciate the raised score.

---

### Official Review · Reviewer_J9Xk · 2025-03-14

**Overall Recommendation:** 3

**Summary:**

This work introduce a method to control the sampling density in diffusion models. The main contribution is using score alignment to scale and control the sampling guidance, which works for both deterministic and stochastic sampling.

The experiments demonstrate that density guidance and its stochastic extension provide fine-grained control over image details.

**Claims And Evidence:**

yes

**Essential References Not Discussed:**

N.A

**Experimental Designs Or Analyses:**

yes.
However, the experiment only demonstrates the side effects of density sampling. I have not observed any positive impact of density sampling on sampling quality.

**Methods And Evaluation Criteria:**

yes

**Other Comments Or Suggestions:**

n.a

**Other Strengths And Weaknesses:**

This paper introduces a density guidance for both deterministic and stochastic processes, enabling precise control over the likelihood (log-density) during the sampling process. Additionally, this work provides solid theoretical foundations that can enhance the community’s understanding of the sampling process. Furthermore, the study uncovers some interesting phenomena: high-density generated results may appear relatively blurry, while lower-likelihood samples introduce more intricate details.

One concern I have is that I have not observed any positive impact of the authors’ proposed solution on existing sampling techniques. While I acknowledge the authors’ contribution and the effectiveness of the density guidance, I would appreciate it if the authors could provide practical advice on how the proposed density guidance might improve current sampling methods.

However, I must still express that I am inclined to accept this work.

**Questions For Authors:**

I hope the authors can offer suggestions on how density guidance could be used to optimize the current sampling process.

**Relation To Broader Scientific Literature:**

This paper may enhance the community’s understanding of the sampling process, and some of the perspectives presented are quite interesting.

**Theoretical Claims:**

yes

---

> ### Author Rebuttal · Authors · 2025-04-01
>
> We would like to thank the Reviewer for their support of our work. Below we address the raised concern.
>
> **File with new figures:** https://anonymous.4open.science/r/DensityGuidance-20E6/Density_guided_sampling___Rebuttal.pdf
>
> > I would appreciate it if the authors could provide practical advice on how the proposed density guidance might improve current sampling methods.
>
> The Density Guidance works on any diffusion model without retraining, or finetuning and requires no extra cost during sampling. We now ran a new experiment and measured the predictions of the generated samples using NIQE [1], which is a metric for image quality assessment reported to correlate strongly with human judgement.
>
> ### Potential applications of detail control
>
> We believe that potential applications of the presented methods are image editing, where the user might want to control the amont of detail in the image. From [2] we also know that highest densities contain cartoon-like images, and thus user can have fine grained control over the spectrum between realistic images and cartoons describing the same scene. People also acknowledge that image generation can be used for “aiding designers in producing striking scenes for video games” [article](https://news.mit.edu/2025/ai-tool-generates-high-quality-images-faster-0321) and we believe that detail control can become an addition to that toolkit.
>
> We also note that there has been interest among the practitioners in explicitly controlling the amount of detail in image generation:
>
> * Modifying Stable Diffusion to generate less detail [Thread](https://www.reddit.com/r/StableDiffusion/comments/1e4spmo/how_to_generate_images_with_less_detail/)
> * Modifying Stable Diffusion to generate more detail  [Thread](https://www.reddit.com/r/comfyui/comments/1866y53/how_do_i_get_more_detail/)
>
> Finally, we would also like to point out that the density-guidance is derived to control the log-density of the generated samples. We know from prior literature [2], that for image data, this correlates with image detail. Perhaps in domains different than images, controlling log-density may be desirable for other purposes. Density Guidance can be used for that as well. Investigation of domains other than image data is out of scope for this work but certainly an interesting direction that we hope this work could pave way for.
>
> We hope our response has addressed the Reviewer's concerns, and that the additional experiments provided in the uploaded file further strengthen your support for our submission, which we hope will be reflected in an updated score.
>
> ---
>
> [1] Mittal et al. "Making a “completely blind” image quality analyzer" (IEEE Signal processing 2012)
>
> [2] Karczewski et al. "Diffusion Models as Cartoonists: The Curious Case of High Density Regions" (ICLR 2025)

---

> > ### Comment · Reviewer_J9Xk · 2025-04-05
> >
> > Thank the author for the reply. While my concerns still seem to exist, I believe that the work's contribution in terms of analysis is still worthy of acceptance.

---

> > > ### Author Response · Authors · 2025-04-05
> > >
> > > Thank you for your follow-up.
> > >
> > > In your initial review, you raised the following concern:
> > > "I would appreciate it if the authors could provide practical advice on how the proposed density guidance might improve current sampling methods."
> > >
> > > In your most recent comment, you mention that "concerns still seem to exist," but it's unclear to us which concerns you're referring to, or why our rebuttal may have fallen short in addressing them. We would genuinely appreciate more clarity, as this would help us better understand how the work could be improved.
> > >
> > > Since the ICML policy this year does not allow us to respond to future comments, we'd like to take this opportunity to clarify and emphasize how we believe we address the concern you raised in your review:
> > >
> > > ### Ease of application to existing models/sampling methods
> > >
> > > In the paper, (line 311 right) we explain how density guidance can easily be implemented as simply as an appropriate rescaling of the score function. This means that for any trained model, **we can apply guidance without any retraining or finetuning, simply by performing regular sampling with a rescaled score function**, regardless of what noise schedule the sampler uses or whether the solver is 1st or 2nd order. We have demonstrated it for
> > > 1. EDM2, which uses PF-ODE with 2nd order Heun solver.
> > > 2. Stable Diffusion, which uses the DDIM solver.
> > > 3. **Now we also added FLUX.1-dev, which is a Flow Matching model** (which is known to be equivalent to diffusion: https://diffusionflow.github.io/), which uses the Euler Solver for sampling.
> > >
> > > This demonstrates that our methods can easily be applied on top of various flow-based models, regardless of how they were trained or what sampling methods they use.
> > >
> > > ### Potential application
> > >
> > > We have also explained how our methods can be useful for applications such as image editing, and also highlighted that we demonstrate how to control log-density. We know that this corresponds to detail control in case of images, but this opens up possibilities of controlling log-density, which may prove useful for other purposes in data other than images.
> > >
> > > We hope that this helped address any remaining concerns you may have and will appreciate if you could reconsider your score.

---

### Official Review · Reviewer_8ij3 · 2025-03-16

**Overall Recommendation:** 4

**Summary:**

The paper proposes a novel method, Density Guidance, to control the level of detail in generated images of flow models. It addresses the observsed mismatch between image likelihood and perceptual quality. The samples of high-likelihood are usually overly smooth, while the low-likelihood ones are more detailed. The author analyze the Prior Guidance adn introduce score alignment. They then propose Density Guidance to enable explicit log-density control by modifying the generative ODE. The model is further extended to stochastic sampling, enabling precise log-density control while allowing controlled variation in structure or fine details. The experimental results demonsrate taht the proposed method can adjust image detail while maintaining the image quality.

### update after rebuttal
The rebuttal has addressed my concerns regarding the perceptual evaluation and ablation study. The authors have provided satisfactory answers about the relationship between perceptual metrics and score alignment, as well as clarified the distinctions between ODE sampling and stochastic sampling approaches. Based on these clarifications, I have decided to increase my score to 4.

**Claims And Evidence:**

The paper make several claims
1. Density Guidance enables explicit log-dencity control. It is justified by a derivation modifying the generative ODE.
2. Score Alignment explains prior guidance. It is supported by a theoretical analysis.

**Essential References Not Discussed:**

It may be better to discuss some papers about perceptual quality metrics and detail-preservation.

**Experimental Designs Or Analyses:**

The experiments effectively validate the proposed method. It conducts the experiments on CIFAR-10 and ImageNet datasets. The proposed method is compared to stable diffusion and edm2. The paper also evaluates the relationship between log-density and perceptual detail.

**Methods And Evaluation Criteria:**

The proposed method is well-justified for controlling image detail in flow models. The use of score alignment to explain prior guidance provide a good insight. The evaluation contrains comparison of generated images, analyses between log-density and perceptual metrics and quantitative evaluation of proposed method, which are appropriate.

**Other Comments Or Suggestions:**

N/A

**Other Strengths And Weaknesses:**

Strengths:
1. Good theoretical contribution. The paper introduces score alignment, which explains the relationship between prior guidance and image detail. This provides a solid theorecial insight for the observation in prior work.
2. The method is well-motivated. The paper proposes density guidance, which enables log-density control insteand of heuristic modification. It can be used in ODE framework of diffusion models and extended to stochastic sampling, which allows controoled variation. The method does not need any additional training.
3. Comprehensive expriments. The paper validate the method on CIFAR-10 and ImageNet. The method is compared to Prior Guidance and demonstrate better control over image detail.

Weaknesses:
1. The user study or perceptual evaluation is missing. LPIPS, FID, SSIM or user study could be added to strengthen the claims.
2. Lack of a comprehensive ablation study. The paper introduces multiple modifications, including score alignment, density guidance, and stochastic density guidance, but it does not conduct a ablation study to evaluate the contributions of each component.

**Questions For Authors:**

1. How does density guidance compare to methods that use explicit perceptual loss functions, such as LIPIS for controlling detail?
2. What is the relationship between preceptual metrics and score alignment?
3. How different does the model perform in ODE sampling the stochastic sampling with Density Guidance?

**Relation To Broader Scientific Literature:**

The work is related to the literature on diffusion models and normalizing flows. It connects well to prior fingdings on the relationship between likelihood and image detail.

**Theoretical Claims:**

The theoretical analysis about score alignment and density guidance looks sound. The authors provide detailed derivations in the appendix.

---

> ### Author Rebuttal · Authors · 2025-04-01
>
> We thank the Reviewer for their positive comments and constructive feedback. We address the raised points below.
>
> **File with new figures:** https://anonymous.4open.science/r/DensityGuidance-20E6/Density_guided_sampling___Rebuttal.pdf
>
> > Discuss some papers about perceptual quality metrics and detail-preservation.
>
> Certainly. In the camera-ready revision we will include a discussion on perceptual quality including the reference-based metrics such as the ones suggested (LPIPS, FID, SSIM), as well as no-reference-based, such as NIQE [3].
>
> > The user study or perceptual evaluation is missing. LPIPS, FID, SSIM or user study could be added to strengthen the claims.
>
> Thank you for this suggestion. The metrics proposed are "Reference-based" metrics, which compare the generated images to reference ones. LPIPS and SSIM score a single image against a single reference image, while FID compares the set of generated images to the set of "real" images. The issue with LPIPS and SSIM is that, for a given generated image, we do not have a corresponding "ground truth" image to compare to. FID would have been more suitable for our use-case, however it is computationally expensive, requiring generating tens of thousands images [1], which was beyond our coomputational budget. It has also been reported to not always agree with human judgement [2].
>
> We thus propose to use NIQE [3], which is a "no-reference" image quality evaluation metric, which is reported to correlate strongly with human judgement. It provides a single number per image, which indicates whether an image has been distorted (a lower number - higher quality). It was used e.g. by [4] to evaluate super-resolution diffusion models.
>
> We evaluated Density and Prior Guidance for EDM2 model, and the now included State-of-the-art model FLUX. In Figure 14 you can see that:
> * For EDM2 model: guided samples can obtained better (lower) NIQE scores than regular samples (gray area);
> * For FLUX model: regular samples already score optimally.
>
> After a visual inspection of optimally scoring guided samples (as measured by NIQE) in Figure 15, we noticed that NIQE actually prefers images with significantly less detail than regular samples. For the FLUX model (Figure 16), there were no perceptual differences between regular samples and best NIQE scoring ones.
>
> > The paper introduces [...] score alignment, density guidance, and stochastic density guidance, but it does not conduct a ablation study to evaluate the contributions of each component.
>
> Thank you for this question. We take this opportunity to clarify:
> * Score alignment (SA) is a novel framework to verify whether a known procedure (Prior Guidance) will be effective in practice
> * (Stochastic) Density Guidance is a novel algorithm proposed by us, which is principled and can be used with any diffusion model (regardless of whether SA holds)
>
> That said, we now performed an extensive evaluation of Prior and Density Guidance, both quantitative and qualitative, including novel models. Please see our response to Reviewer ppbq for more details.
>
> > How does density guidance compare to methods that use explicit perceptual loss functions, such as LIPIS for controlling detail?
>
> The difference between Density Guidance (DG) and models trained with perceptual loss functions is two-fold. First, DG can be used, without any finetuning or retraining, and for no extra cost, on models which were trained without any perceptual losses. Second, it provides a way to control the generations. One can generate images with either high or low level of detail, depending on the use-case. Models trained with perceptual losses do not have that capability.
>
> > What is the relationship between preceptual metrics and score alignment?
>
> Score Alignment guarantees that Prior Guidance effectively changes log-density in deterministic sampling. We show in Figure 14 how that can impact the perceptual metrics.
>
> > How different does the model perform in ODE sampling the stochastic sampling with Density Guidance?
>
> Please see our response to Reviewer ppbq for the details on the comparison on different modes of sampling.
>
> We thank the Reviewer again for their useful suggestions that helped improve our work. We hope that our clarifications and additional experiments addressed all concerns and ask for a reconsideration of the score.
>
> ---
>
> [1] Heusel et al. "GANs trained by a two time-scale update rule converge to a local Nash equilibrium." (NeurIPS 2017)
>
> [2] Liu et al. "An improved evaluation framework for generative adversarial networks." (arXiv 2018)
>
> [3] Mittal et al. "Making a “completely blind” image quality analyzer" (IEEE Signal processing 2012)
>
> [4] Sami et al. "HF-Diff: High-Frequency Perceptual Loss and Distribution Matching for One-Step Diffusion-Based Image Super-Resolution." (arXiv 2024)

---

### Decision · Program_Chairs · 2025-05-01

**Decision:**

Accept (poster)

**Comment:**

This paper introduces a method for controlling the level of detail in images generated by flow-based models. It first analyzes the phenomenon where high-likelihood samples appear overly smooth, and low-likelihood ones are more detailed. They explain this with a new condition called score alignment, which ensures that scaling the latent noise vector reliably affects the output image's log-likelihood. Building on this, they modify the generative ODE to explicitly steer samples toward a target likelihood, and further extend it to stochastic sampling. Their approach enables fine-grained control over image detail without sacrificing sample quality.

The paper received 2 weak accept, and 2 accept. All the reviewers acknowledged that this paper is well-motivated, with good theoretical contribution. One reviewer raised concerns for validation and experiment design, who later was satisfied with the rebuttal. While the authors' initial response to reviewer `J9Xk`’s question on practical guidance was partial, the ACs confirmed that the subsequent clarification adequately addressed how Density Guidance can be used to control fine-grained detail during sampling.

Given the paper's solid quality and positive feedback, ACs agreed on acceptance.